# Ubiquitome profiling reveals a regulatory pattern of UPL3 with UBP12 on metabolic-leaf senescence

Wei Lan*, Weibo Ma*, Shuai Zheng, Yuhao Qiu, Han Zhang, Haisen Lu, Yu Zhang, Ying Miao

The HECT-type UPL3 ligase plays critical roles in plant development and stress protection, but understanding of its regulation remains limited. Here, the multi-omics analyses of ubiquitinated proteins in *upl3* mutants were performed. A landscape of UPL3-dependent ubiquitinated proteins is constructed: Preferential ubiquitination of proteins related to carbon fixation represented the largest set of proteins with increased ubiquitination in the *upl3* plant, including most of carbohydrate metabolic enzymes, BRM, and variant histone, whereas a small set of proteins with reduced ubiquitination caused by the *upl3* mutation were linked to cysteine/methionine synthesis, as well as hexokinase 1 (HXK1) and phosphoenolpyruvate carboxylase 2 (PPC2). Notably, ubiquitin hydrolase 12 (UBP12), BRM, HXK1, and PPC2 were identified as the UPL3-interacting partners in vivo and in vitro. Characterization of *brm*, *upl3*, *ppc2*, *gin2*, and *ubp12* mutant plants and proteomic and transcriptomic analysis suggested that UPL3 fine-tunes carbohydrate metabolism, mediating cellular senescence by interacting with UBP12, BRM, HXK1, and PPC2. Our results highlight a regulatory pattern of UPL3 with UBP12 as a hub of regulator on proteolysis-independent regulation and proteolysis-dependent degradation.

## Introduction

Cell senescence, including developmental senescence and stress-induced cell senescence, is triggered by internal and external factors and often involves degradation and remobilization of cellular components. During plant senescence, a visible change of leaf yellowing is an indication of chloroplast damage and chlorophyll degradation. At the molecular level, catabolism of macromolecules is a major event in senescent cells, especially involving proteolysis. It has been documented that during *Arabidopsis* leaf senescence, alterations in transcriptional regulation, histone-associated epigenetic processes, posttranslational modification (PTM), and macromolecule/organelle degradation are genetically determined and developmentally programmed (Buchanan-Wollaston et al, 2005; Woodson et al, 2015; Yolcu et al, 2017). Pathways for protein degradation include the 26S proteasome, the organelle degradation, autophagy processes, and the monoubiquitination- or short ubiquitin chain–dependent proteinases (Vierstra, 2009; Miller et al, 2010; Berndsen & Wolberger, 2014; Lan & Miao, 2019). The targeting for 26S proteasome degradation is a sequential process that starts with the ubiquitin activation by the E1 (ubiquitin-activating) enzyme in an ATP-dependent manner. The activated ubiquitin is then transferred from the E1 to the E2 (ubiquitin-conjugating) enzyme that acts as an intermediate. Finally, the E3 (ubiquitin ligase) enzyme mediates the deposition of the activated ubiquitin to, normally, a lysine residue of the target protein. Depending on the E3 type, protein ubiquitination may be a direct or indirect process (Berndsen & Wolberger, 2014; Zheng & Shabek, 2017).

The HECT E3 ligase family contains seven members (UPL1-UPL7) in *Arabidopsis*, which plays important roles in protein fate and protein function during the senescence process (Lan & Miao, 2019). Although animal studies have revealed diverse mechanisms in the functions and regulation of HECT E3s, their plant counterparts are less explored. So far, the *Arabidopsis* UPL3 and UPL5 are known to be involved in plant development and response to stress (Downes et al, 2003; Miao & Zentgraf, 2010; Patra et al, 2013; Bensussan et al, 2015; Furniss et al, 2018; Miller et al, 2019). Although the *upl5* mutant shows a premature aging phenotype, UPL3 functions in trichome, vascular, and seed development, as well as in immune response. The UPL5 protein is able to target the transcription factor WRKY53 for ubiquitination and degradation, playing an antagonist role in leaf senescence (Miao et al, 2004; Miao & Zentgraf, 2010). UPL3 may target GLABROUS 3 (GL3) and ENHANCER OF GL3 (EGL3), two bHLH transcription factors, and positively regulate trichome development and flavonoid biosynthesis in *Arabidopsis* (Patra et al, 2013). The *upl3* mutant shows larger stem diameter than that of the WT plant, indicating a role of UPL3 in regulation of vascular development (Bensussan et al, 2015). UPL3 is required for development of plant immunity (Furniss et al, 2018). Recently, Miller et al (2019) showed that UPL3 controls the protein stability of LEAFY COTYLEDON2 (LEC2), a key transcriptional regulator of seed maturation, and regulates the seed size and crop yields (Miller et al, 2019). Thus, UPL3 emerges as a critical player with pleiotropy traits in

Fujian Provincial Key Laboratory of Plant Functional Biology, College of Life Sciences, Fujian Agriculture and Forestry University, Fuzhou, China

Correspondence: ymiao@fafu.edu.cn
*Wei Lan and Weibo Ma contributed equally to this work.

*Arabidopsis thaliana*. However, the underlying mechanisms are still limited understood.

In this study, we conducted globally proteomic and ubiquitomic analyses by using a label-free mass spectrometry-based analysis of protein ubiquitination with the di-Gly-Lys–remnant antibody enrichment approach in the *upl3* mutant relative to the wild type to construct a landscape of UPL3-dependent ubiquitylated proteins, and analysis of ubiquitin footprint provides direct evidence of the ubiquitination of proposed target proteins. GFP nanotrap–mass spectrometry and yeast two-hybrid assay further confirmed direct UPL3 targets. Further UPL3-interacting proteins combined with their mutants' phenotyping and analysis of transcriptome dataset suggested that UPL3 targeted deubiquitination enzymes (DUBs) and acted as regulators impacting on metabolism-related cell senescence.

# Results

### Mutation of *UPL3* affects plant development

To address more defined roles of UPL3, we systematically analyze the phenotype of *upl3* lines; two knockout lines (*upl3-1*, *upl3-3*), two overexpressing lines (*oeUPL3-24*, *oeUPL3-39*), and two complementation lines (*comUPL3-1*, *comUPL3-2*) were produced and confirmed at the transcriptional and protein levels (Figs 1A and S1). Plants of the *upl3* lines exhibited apparently downward-curled leaves at the late stage and a 2-wk delay in bolting and leaf senescence, compared with the wild-type (WT) plant (Figs 1B–D and S2C). In contrast, the overexpressing UPL3 lines displayed premature leaf aging and 1-wk earlier bolting (Figs 1B–D and S2C), and lower number of rosette leaves are observed in *UPL3* overexpressing plants (Fig 1D). Complementation by the full-length UPL3 rescued the *upl3* mutants' phenotypes (Fig 1B–D). Consistently, chlorophyll content (Fig 1E), photosystem II fluorescent activity (Fv/Fm) (Fig 1G), and green leaves/yellow leaves ratio (Fig 1F) increased significantly in the *upl3* lines and decreased in the *oeUPL3* line, compared with the WT. And the transcript level of early senescence marker gene (*WRKY53*) and floral transition marker gene (*FT*) declined significantly in the *upl3* lines and increased significantly in the *oeUPL3* line (Fig S2D). Thus, UPL3 functions in organ development, aging, and flowering by accelerating leaf senescence, under natural developmental conditions.

### Loss of UPL3 induces global ubiquitin enrichment in 6-wk-old *upl3* plants

UPL3 was highly expressed in the senescent leaf (after 6 wk) (Winter et al, 2007; Fig S2A and B); therefore, we performed a label-free mass spectrometry (MS)–based analysis of protein ubiquitination using the di-Gly-Lys–remnant antibody enrichment approach using 6-wk-old *upl3* and WT plants (Figs 2A and S3). Identified proteins based on their tandem mass spectra matching against the UniProt *A. thaliana* Columbia database (current total of 39,211 reads) using the MaxQuant software are listed in Supplemental Data 1. Label-free quantification, with a false discovery rate (FDR) adjusted to <0.01

and a minimum score for modified peptides set as >40, resulted in a set of over 1,310 potential ubiquitinated targets. This was further refined to a subset of 1,155 targets by at least two PTMs (Fig 2B and Supplemental Data 2).

To identify the alteration of ubiquitin conjugates associated with the *UPL3* mutation, a fold-change greater than 1.2 or less than 1/1.2 and *P*-value < 0.05 of two replicates were used to filter conjugate targets in the library, whose ubiquitination was up-regulated or down-regulated. All the differentially ubiquitin conjugates (DUCs) data in *upl3*/WT (|FC| > 1.2, *P*-value < 0.05) were shown in Fig 2B and Supplemental Data 2. Among them, ubiquitination of 545 sites (356 proteins) was found to be up-regulated, and ubiquitination of 198 sites (189 proteins) was down-regulated in the *upl3*, compared with the WT plant (Fig 2B). These unexpected results were verified by global ubiquitination immunodetection using an anti-polyubiquitin antibody, in which deletion of *UPL3* led to enhanced signals of global levels of protein ubiquitination, whereas loss of its homolog *UPL5* did not, as a control (Fig 2C, Miao & Zentgraf, 2010). Over-expression of *UPL3* retained comparable global levels of protein ubiquitination with the WT (Fig 2C). Indeed, more proteins were found in the up-regulated category than in the down-regulated set, in term of modified protein sites (Fig 2D). Specifically, 188 conjugates had enhanced abundance over WT fold-change by greater than 1.5-fold, whereas the levels of only 79 conjugates were reduced fold-change by less than 1/1.5-fold (0.67) (Fig 2D). Fold-change levels relative to the statistic *P*-value of individual ubiquitinated site were plotted to give more details on the dataset (Fig 2E). Of 267 DECs (|FC| > 1.5) was the fact that ubiquitin-conjugating enzyme 35 (UBC35) and ubiquitin-activating enzyme 1 (UBA1) were among the targets with significantly enhanced ubiquitination levels (Fig 2E), and RPN10, a known target of UPL3 (Furniss et al, 2018), was appeared in dataset of ubiquitome with a down-regulated ubiquitination level, although several conjugates including HXK1, PPC2, VAMP714 etc. were indicated in *P*-value > 0.05 (yellow dots) of two replicates; they were both significantly down-regulated in two replicates (Supplemental Data 2), indicating that the quality of ubiquitomic datasets were properly sound. Together, this surprising rise in protein ubiquitination caused by loss of *UPL3* perhaps means that UPL3 has function on deubiquitinating pathway or UPL3 promotes deubiquitinases (DUBs) function.

### Differential ubiquitomic enrichment reveals that metabolic enzymes are among the processes significantly affected by UPL3

GO term enrichment showed that the molecular functions of assembled ubiquitin conjugates were related to protein–protein interactions (47%), catalytic activity (34%), transport activity (7%), structural molecule activity (5%), and other (7%). For the DUCs (|FC| > 1.2, *P*-value < 0.05) of *upl3* relative to WT, the protein–protein interaction function enrichment was slightly increased to 49% (Fig 3A), implying a major involvement in molecular interaction for the UPL3-regulated targets. Further assignment to biological processes showed that the DUCs included mainly proteins involved in the response to metal ion stresses and enzymes related to carbon metabolism and nucleotide metabolism (Fig S4). Hence, it is plausible that UPL3 maintains the ubiquitination status of enzymes and regulatory factors to fine-tune cellular metabolism.

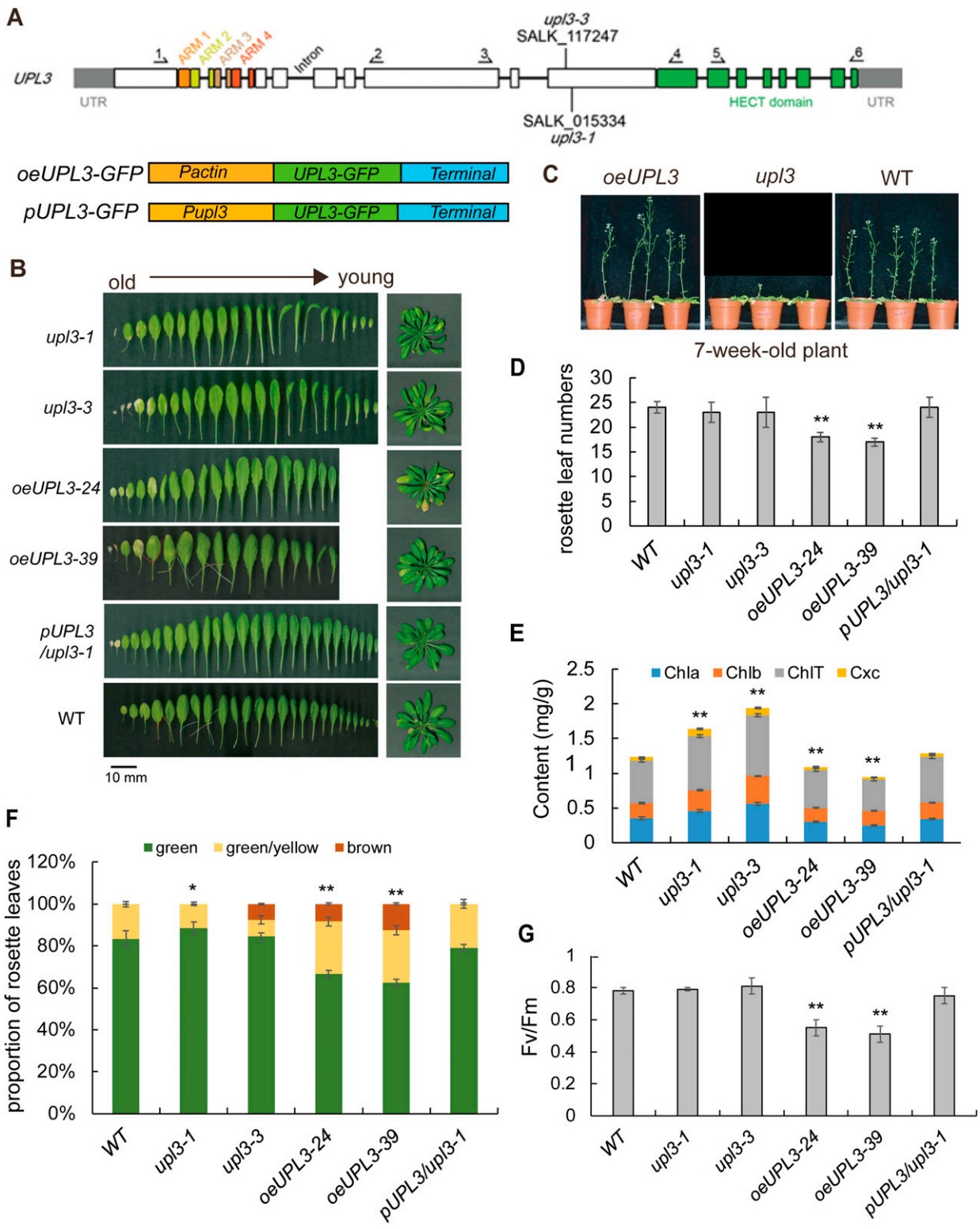

**Figure 1. Phenotype of representative 6-wk-old wild-type and *upl3* plants.**
**(A)** Organization of the *UPL3* gene and plasmid constructs for transgenic plants. **(B)** Phenotype of representative 6-wk-old wild-type, *upl3*, and *oeUPL3* transgenic plants, as well as complemented line (*pUPL3/upl3-1*). Leaf senescence and curled leaves are indicated. Bars = 10 mm. **(C)** Representative images showing the bolting of plants in the seventh week after germination. **(D)** Rosette leaf numbers of the wild-type, *upl3-1*, *oeUPL3-24*, and *pUPL3/upl3-1* plants (n > 20), indicating flowering time. **(E)** Chlorophyll content and carotenoid content measured in the rosettes of 6-wk-old wild-type, *upl3-1*, and *oeUPL3-24* plants. Chla, chlorophyll a; Chlb, chlorophyll b; ChlT, total chlorophyll; Cxc, carotenoid. **(F)** Proportion of green and yellow leaves of whole rosette of 8-wk-old plants (n = 12). Error bars represent the SD of six biological

This notion was further validated via KEGG pathway analysis (Figs 3B and S4). A large subset of the DUCs was enzymes related to biosynthesis of secondary metabolites, carbon fixation, carbon metabolism, and amino acid metabolism. The ubiquitinated forms of the Calvin–Benson enzymes, such as ribulose-1,5-$P_2$-carboxylase (RuBisCO), phosphoglycerate kinase (PGK), GAPDH, and the aldo-keto reductase family members (ALDOs), as well as the CAM enzymes malate dehydrogenases MDH1 and MAEB homolog, were enriched in the *upl3* plants (Fig 3C). On the other hand, a small subset of declined ubiquitin conjugates was cysteine and methionine metabolism–related enzymes, such as 5-methyltetrahydropteroyl-triglutamate, homocysteine methyltransferase 2 (MS2), S-adenosylmethionine synthase 1 (SAM1), cysteine synthase 1 (OASA1), methionine aminotransferase (BCAT4), SNARE-like superfamily protein (YKT61), and vesicle-associated membrane protein 714 (VAMP714) in the *upl3* plants (Fig 3D and Supplemental Data 2). Interestingly, carbon metabolism was also represented by a reduction in ubiquitin conjugates of phosphoenolpyruvate carboxylase (PPC2) and hexokinase 1 (HXK1) in the *upl3* background (Fig 3C and Supplemental Data 2).

Our *Arabidopsis* ubiquitome proteins can be classified as cytoplasmic (36%), nuclear (20%), chloroplast (20%), membrane (14%), and others (10%). Among these, the percent ratio of the UPL3-related DUCs (|FC| > 1.2, *P*-value < 0.05) was higher in the cytoplasmic and the nuclear sections (Figs 3E and S4). The Cytoscape protein–protein interaction network of the 545 UPL3-related DUCs generated distinctly clustered interaction nodes (Fig S5) (Shannon et al, 2003). The UPL3-regulated conjugates were dispersed throughout the network, suggesting that UPL3 was likely involved in broad control of stress or stimulate response (Fig S5). UPL3 fused GFP transformed transiently in tobacco leaves showed that UPL3 was clearly localized in the nucleus (Fig 3F). Therefore, nuclear proteins such as those regulatory functioning in chromatin remodeling, chromosome regulation, and transcriptional complexes related to stresses are likely primary UPL3-dependent targets.

### The UPL3-dependent conjugates are enriched in histone H1/H5 and stress-related protein domains with a noncanonical lysine pattern

Scanning protein domains of the DUCs using InterProScan identified several UPL3-ubiquitinated (reduced ubiquitination in the *upl3* background) protein domains including the histone H1/H5 domain, jacalin-like lectin domain, GST domain, S15/NS1 RNS binding domain, and heavy-metal–associated domain (Fig 4A blue), whereas the UPL3-regulated ubiquitin conjugates (enhanced ubiquitination in the *upl3* background) contained leucine-rich repeat, histone H2A/H2B/H3, ribosomal protein S5, double-stranded RNA-binding domain, and histone fold domains (Fig 4A orange). For protein domains of the individual DUC within each fold-change range (Q1, Q2, Q3, and Q4), the results were shown in Fig S4 and Supplemental Data 2. Again, it was noted that protein domains associated with RNA binding, protein translation, and

amino acid synthesis–related proteins were overrepresented in proteins showing reduced ubiquitination in the *upl3* plants, followed by heavy-metal–associated domain and remorin- and jacalin-like lectin domain of abiotic (metal)/biotic stress–responsive proteins. In contrast, protein domains associated with protein binding, transport ATPase, ubiquitin carboxyl-terminal hydrolase, and metabolism enzyme were the most significant domains in proteins showing enhanced ubiquitination in the *upl3* plants, followed by domains contained in the ubiquitination/26S proteasome system (UPS) regulatory complex (Fig S6 and Supplemental Data 2). It suggested that proteins containing heavy-metal–associated domains, GST domains, jacalin-like lectin domains, and histone H1/H5 domains were likely UPL3-ubiquitinated conjugates. Correspondingly, UPL3-regulated conjugates contain chromatin remodeling ATPase BRM, histone H2A/H2B/H3, histone fold, and SPK1/BTB/POZ–binding domain, most of which have functions related to protein binding activity and gene transcriptional control (Fig S6). In this scenario, UPL3-regulated conjugates in the nucleus are likely mediators of chromatin accessibility and transcriptional processes that control downstream gene expression.

Ubiquitin can be attached to substrate proteins as a single moiety or polymeric chains and adopt distinct conformations and lead to different functions in cells (Komander & Rape, 2012). To get insight ubiquitinated lysine site pattern of UPL3, the H89R substitution in the tagged ubiquitinated assay was used here to enable the detection of ubiquitination sites ("footprints") and identify a consensus ubiquitin attachment sequence (Xu et al, 2010). By scanning all generated datasets, we identified 2,778 ubiquitinated sites in total (Supplemental Data 1). Among these, 2,359 sites were quantified, 1,641 of the 2,359 sites were in the *upl3* plants, and 414 sites were differentially displayed relative to WT plant with a cutoff $\log_2 FC$ of 1.5.

We identified 110 ubiquitinated modification sites on 77 differentially ubiquitinated proteins in *upl3*/WT ($\log_2 FC > 2$, *P*-value < 0.05) (Supplemental Data 2). Motif analysis around the modified lysine using MEME identified a consensus ubiquitin attachment sequence in 44 of the 110 sites (Fig 4B) that strongly matched the c-K-x-E/D/G ubiquitination motif (where c and x represent a hydrophobic and any amino acid, respectively), which was a prevalent motif in yeast and animal ubiquitinated targets (Xu et al, 2010). However, the remaining 66 sites (60%) were unrelated to this motif, indicating that noncanonical sites were also common. In addition, referring to the GPS-SUMO algorithm (Zhao et al, 2014), one or more copies of this consensus sequence were detected in ubiquitinated targets accounting for 74%, 66%, and 59% of the three enriched ubiquitination categories, the UPL3-ubiquitinated, abundant, and total ubiquitinated proteins, respectively. Among the 44 sites with a consensus sequence, of the 110 sites, 14 of the 21 mapped attachment sites on 18 UPL3–up-regulated targets belonged to the canonical c-K-x-E/D/G motif, with the remainder had alternative sequences (Supplemental Data 3), including GAPC1, GAPC2, CASA1, NADP-ME2, RuBisCO, AAT1, FBAB, PGK3, MDH1, GLO1, and SHM4 (Fig 4C). Specifically, the ubiquitinated sites identified here for the

---

replicates. Asterisks denote statistically significant differences from the WT, calculated using *t* test: \*P < 0.05; \*\*P < 0.01; and \*\*\*P < 0.001. **(G)** Photosystem II fluorescence activity (Fv/Fm) of the fifth rosette leaf of 6-wk-old plants (n = 9).

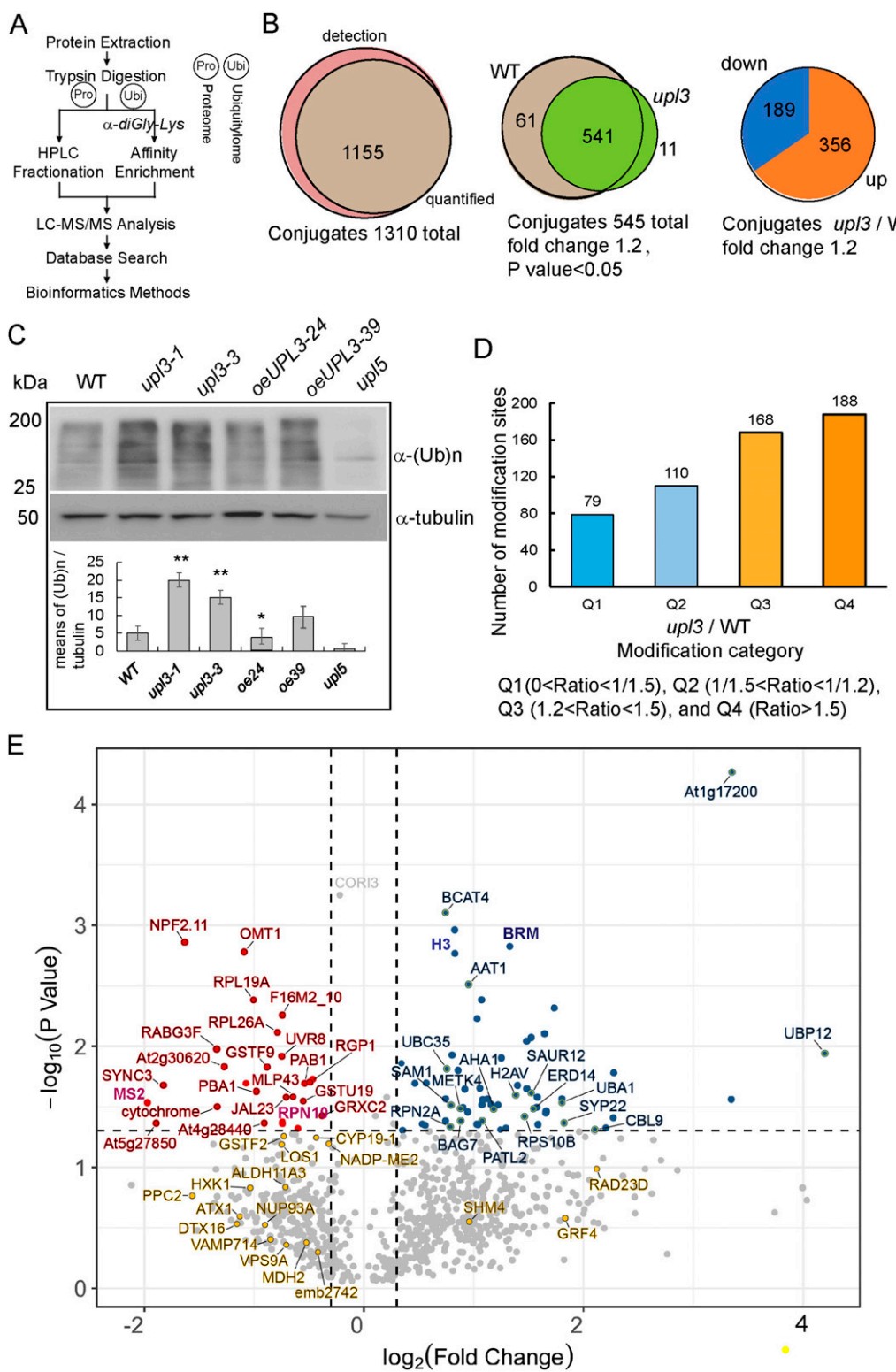

**Figure 2. Identification of ubiquitin conjugates in 6-wk-old *upl3* plants compared with the wild type.**
**(A)** A label-free mass spectrometry–based analysis procedure of protein ubiquitination using K-epsilon-GG remnant antibody enrichment approach. **(B)** Venn diagrams showing the enrichment of differentially ubiquitinated proteins in the *upl3* background. **(C)** Immunodetection of global ubiquitinated proteins in plants of WT and mutants using an antibody against ubiquitin. The *upl5* plant is included for a comparison. WB of β-tubulin is used as protein-loading controls. The gray intensity of three replicates was calculated by ImageJ. **(D)** Distribution of enriched proteins within differential fold-change levels by *upl3* relative to WT. **(E)** Volcano plot of individual ubiquitination site showing their *P*-value and the log$_2$FC. Dark-gray points are conjugates considered to be "abundant" by their detection in three biological replicates in

UPL3-ubiquitinated targets, namely, ALDH, OASA1, HXK1, and PPC2, were within a noncanonical x-A-K-x- motif or x-K-A-x motif (Fig 4C).

### Identifying the potential ubiquitylated targets in UPL3-bound proteins

To evaluate the direct connection between UPL3 and conjugates, we assessed protein–protein interaction using a GFP nanotrap–assisted pulldown-MS assay (Fig S7), in which the GFP-tagged UPL3 was a bait. To the end, the rosette leaves of 6-wk-old stable transgenic plants expressing either the UPL3-GFP or the control GFP driven by the *ACTIN3* promoter were used for total protein isolation, purification, and immunodetection (Fig 5A and B). The trypsin-digested proteins were subjected to mass spectrometry analysis with high-energy collisional dissociation quantum efficiency mass spectrometry (QE-MS). Tandem mass spectra were searched against UniProt *A. thaliana* Columbia (89247_20181227) database via Mascot2.2 software. A total of 81 putative proteins were identified after subtracting the GFP control resulted from the UPL3-GFP candidate list (Supplemental Data 4). With these putative UPL3-interacting patterns, we identified 29 (|FC| > 1.5, *P*-value < 0.05) or 11 under more stringent conditions (|FC| > 1.5, *P*-value < 0.01) overlapping proteins in the UPL3-ubiquitome DUCs (Fig 5C).

These 29 proteins were clustered in four categories based on the KEGG pathway database. Consistently, proteins involved in the carbon fixation pathway have enhanced ubiquitination in the *upl3* mutant, whereas those in the inositol-1,4,5-trisphosphate-3-kinase (IP3K) signaling pathway have reduced ubiquitination in the *upl3* plants versus WT (Fig 5D and E), which included six proteins, namely, ABCG36, MS2, PPC2, LOS1, HXK1, and AT3G63160 (Fig 5E, fold-change depicted in blue). 14 other interacting candidates showing *upl3* mutation–enhanced ubiquitination in the *upl3* background were H2AXb, RPS2B, PHOT1, ERD14, the SWI-SNF chromatin-remodeling ATPase BRAHMA (BRM), and SWIS3C, as well as carbon metabolism–related enzymes (Fig 5E, fold-change depicted in orange). Notably, UBP12, UBP13, and UBP26 are also among the UPL3-interacting candidates (Fig 5E). UBP12 is an ubiquitin hydrolase, with a demonstrated deubiquitination activity in vitro and localization both in the cytoplasm and the nucleus (Derkacheva et al, 2016; Kralemann et al, 2020).

### UPL3 interacting with UBP12, HXK1, and PPC2 are formed a complex

To confirm these putative UPL3 interacting partners, we carried out a yeast two-hybrid assay by selecting 16 candidates. The self-interacting N-terminal fragment (470 aa) of UPL3 containing armadillo repeats was shown interacting with most of the selected candidates except for UPL5 and PHOT1; however, the full-length UPL3 only showed a strong interaction with UBP12 and very weak interaction with BRM, HXK1, PPC2, and UBC35 (Figs 6A and B and S8A), but the full-length UPL3 bait was not able to interact with its N-terminus (Fig 6B). Next, the interaction of UPL3 with UBP12 was confirmed using BD-UBP12 as a bait, which in turn showed a weak interaction with PPC2 and BRM, but not HXK1 (Fig 6C).

We further examined the interaction of UPL3 or UBP12 and its interacting partners by bimolecular fluorescence complementation (BiFC) assays and coimmunoprecipitation (CoIP). UPL3 or UBP12 and partners were fused to the YN vector (*pCAMBIA1300-N1-YFPN*) and YC vector (*pCAMBIA1300-N1-YFPC*), respectively, and then were cotransformed to *Arabidopsis* epidermal cells by Agrobacterium-mediated injection. GFP signals in the nucleus were strongly observed in UPL3-GFPn and its interacting partners BRM-GFPc, HXK1-GFPc, PPC2-GFPc, and UBP12-GFPc-co-delivered *Arabidopsis* epidermal cells (Figs 6D and S8B), as well as in UBP12-GFPn and BRM-GFPc and PPC2-GFPc but not in UBP12-GFPn- and HXK1-GFPc-co-delivered *Arabidopsis* epidermal cells (Fig 6E); the empty vector was used a negative control (Fig S8B). Next, we detected the interaction of UPL3 with UBP12 or HXK1 or PPC2 in vivo by Co-IP with the *oeUPL3-GFP* or *oeUBP12-GFP* transgenic plants. An anti-GFP antibody was used for immunoprecipitation and anti-HXK1, anti-BRM, and anti-PPC2 antibodies were then used for IP Western-blotting detection. UPL3-GFP was coimmunoprecipitated with endogenous HXK1 and PPC2 in 6-wk-old plants, and UBP12-GFP was coimmunoprecipitated with PPC2 but not HXK1 (Figs S8C and 6F), indicating that UPL3 is able to interact with HXK1 and PPC2; UBP12 is able to interact with PPC2, and thus, UPL3, UBP12, BRM, PPC2, and HXK1 may form a complex.

### The UPL3 and UBP12 affect HXK1 and PP2C protein stability and carbohydrate contents

To examine whether UPL3 altered the protein level of the bound candidates of UPL3 between *upl3* and WT, to the end, we compared the total proteomes of 6-wk-old wild-type and the *upl3* plants by tandem MS using the precursor ion intensity of the MS1 scans for quantification. Altogether, 3,557 *Arabidopsis* proteins could be reproducibly identified and quantified in both samples by our liquid chromatography–mass spectrometry (LC–MS) regime analyzed in triplicate (Fig 7A and Supplemental Data 5). 492 proteins were up-regulated, and 236 proteins were down-regulated in differentially expressed proteins (DEPs, |FC| > 1.5, *P*-value < 0.05) of proteome dataset from *upl3*/WT (Fig S9 and Supplemental Data 5). When above DUCs (|FC| > 1.5, *P*-value < 0.05) was normalized to DEPs (|FC| > 1.5, *P*-value < 0.05), 105 ubiquitination sites (74 proteins) were enriched and only seven sites (five proteins) were down-regulated (Fig 7B) from *upl3*/WT. Among them, several UPL3-bound proteins such as UBC35, UBC7, BRM, UBP12, and UPL13, as well as other UBP members (e.g., UBP1C, UBP6, UBP26) were slightly up-regulated 1–1.5 fold in the *upl3* mutant relative to WT; the rest UPL3-bound proteins

either background (*upl3* and/or the wild type). Proteins with a significant decrease or increase in ubiquitination in the *upl3* mutant compared with the wild type (*P*-value < 0.05) are highlighted in red and blue, respectively. Ubiquitination targets identified in all wild-type biological replicates and never or only once in the *upl3* mutant but that were above the significance threshold of *P*-value > 0.05 are in yellow. The dashed line represents the theoretical situation, where conjugate abundance in the wild type and *upl3* is equal. The horizontal dashed line highlights a *P*-value = 0.05. The vertical dashed lines highlight a 1.2-fold (log$_2$FC = 0.26) increase or decrease. Source data are available for this figure.

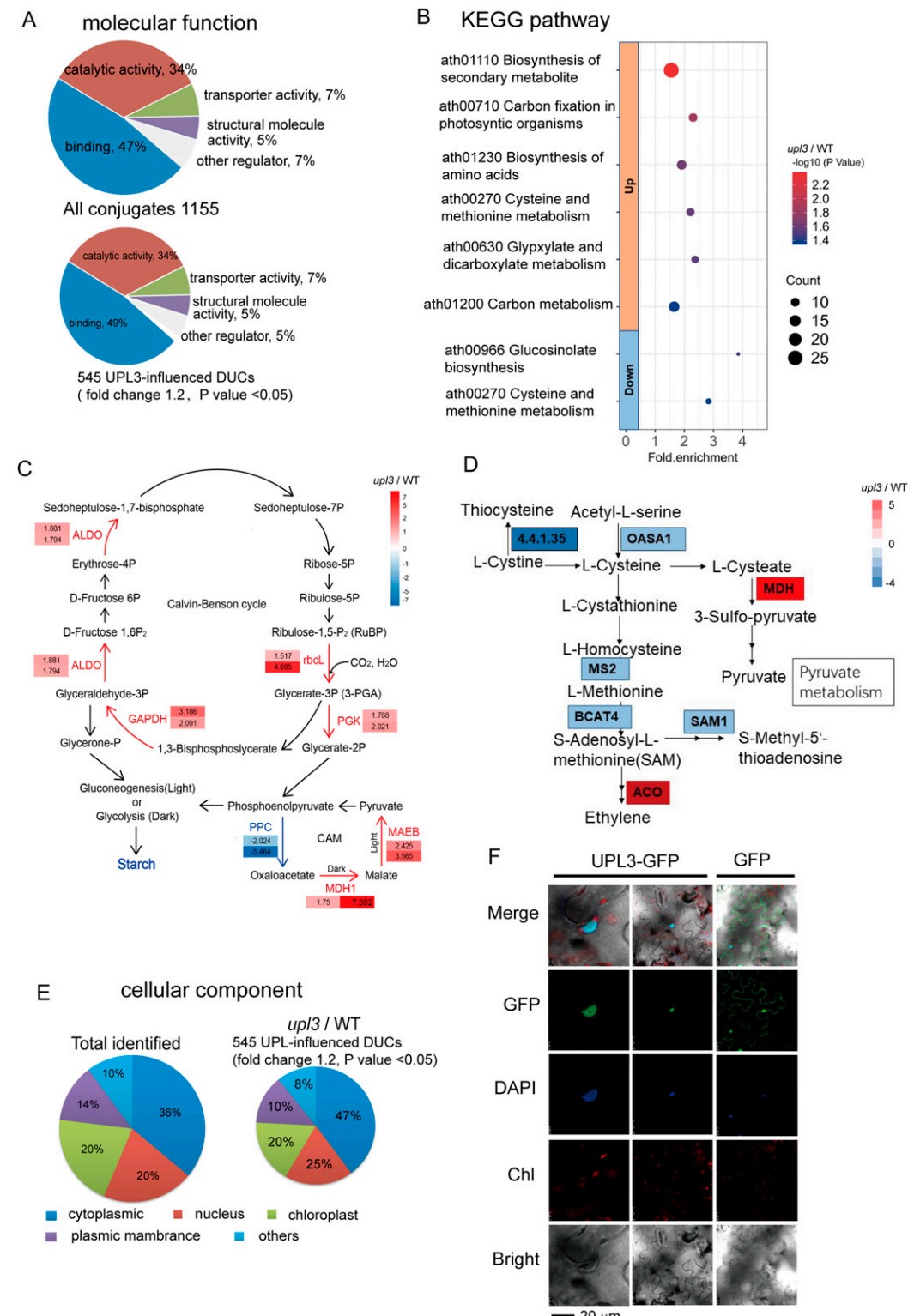

**Figure 3. GO term and KEGG pathway enrichments for the UPL3-associated ubiquitin conjugates (|FC| > 1.2, *P*-value < 0.05).**
**(A)** GO term enrichment in categories of molecular functions of UPL3-associated ubiquitin conjugates. **(B)** Bubble plots showing the KEGG pathway of UPL3-associated ubiquitin conjugate differentially ubiquitin conjugates. **(C)** The UPL3-regulated ubiquitin conjugates of the *upl3*/WT mapped in the carbon fixation pathway. **(D)** The UPL3-ubiquitinated conjugates of the *upl3*/WT mapped in cysteine and methionine metabolism pathway. **(E)** Distribution of UPL3-associated differentially ubiquitin conjugates associated with cellular components. **(F)** Subcellular localization of UPL3 in the nucleus of *Arabidopsis* leaves transiently expressing a C-terminal GFP fusion. The GFP-alone expressing leaves sample is used as the control. The nucleus is counter-stained with DAPI. Bars = 20 μm.

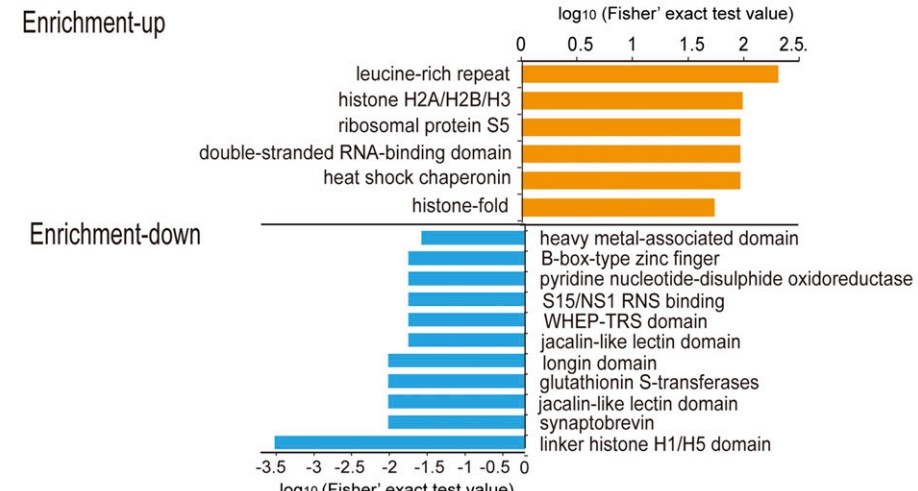

**A**

Enrichment-up

log₁₀ (Fisher' exact test value)

(bar chart)
- leucine-rich repeat
- histone H2A/H2B/H3
- ribosomal protein S5
- double-stranded RNA-binding domain
- heat shock chaperonin
- histone-fold

Enrichment-down
- heavy metal-associated domain
- B-box-type zinc finger
- pyridine nucleotide-disulphide oxidoreductase
- S15/NS1 RNS binding
- WHEP-TRS domain
- jacalin-like lectin domain
- longin domain
- glutathionin S-transferases
- jacalin-like lectin domain
- synaptobrevin
- linker histone H1/H5 domain

log₁₀ (Fisher' exact test value)

**B**

| Motif Logo | Motif | Motif Score | Fold Increase |
|---|---|---|---|
| (logo) | ⟨xxxxxxE_K_xxxxxx⟩ | 9.24 | 1.6 |
| (logo) | xxxxxxxxx_K_xxAxxx: | 7.31 | 1.7 |
| (logo) | xxxxxxxxx_K_Exxxxx: | 6.20 | 1.6 |
| (logo) | ⟨xxxxxxxx_K_Gxxxxx: | 7.05 | 1.8 |

**C**

| Protein Accession | Protein Name | Modified Sequence | Motif | Fold Changed R1 | Fold Changed R2 |
|---|---|---|---|---|---|
| P25858 | GAPC2  K(76) | TLLFGEK(1)PVTVFGIR | ......EK...... | 3.19 | 2.09 |
| Q9FX54 | GAPC1  K(76) | TLLFGEK(1)PVTVFGIR | ......EK...... | 3.19 | 2.09 |
| P47998 | OASA1  K(313) | K(1)EAEAMTFEA | ......KE...... | 2.82 | 1.52 |
| P10798 | RBCS-3B  K(147) | K(1)EYPGAFIR | ......KE...... | 1.9 | 1.58 |
| Q9LYG3 | NADP-ME2  K(372) | IWLVDSK(1)GLIVSSR | ......KG...... | 2.42 | 3.56 |
| Q9LRR9 | GLO1  K(230) | LPILVK(1)GVLTGEDAR | ......KG...... | 1.71 | 1.51 |
| Q8S4Y1 | AAT  K(48) | LGSLAIAAALK(1)R | ......A...K...... | 1.9 | 1.97 |
| Q9LF98 | FBA8  K(38) | GILAADESTGTIGK(1)R | ......A...K...... | 1.88 | 1.79 |
| Q9SAJ4 | PGK3  K(78) | YSLK(1)PLVPR | | 1.79 | 2.02 |
| P93819 | MDH1  K(105) | DVMSK(1)NVSIYK | | 1.75 | 7.3 |
| O23254 | SHM4  K(424) | AVTLTLDIQK(1)TYGK | ......A...K...... | 1.55 | 2.34 |
| O03042 | RBCL  K(146) | IPPAYTK(1)TFQGPPHGIQVER | ......A...K...... | 1.52 | 4.68 |
| F4INS6 | ALDH11A3 K(108) | EIAK(0.04)PAK(0.9)DSVTEVVR | ......AK...... | -1.68 | -1.6 |
| P47998 | OASA1  K(191) | IDGFVSGIGTGGTTTGAGK(1)YLK | ......A...K...... | -1.68 | -1.5 |
| Q42525 | HXK1  K(77) | QVADAMTVEMHAGLASDGGSK(1)LK | ......A...K...... | -1.75 | -2.47 |
| Q5GM68 | PPC2  K(622) | VAK(1)EYGVK(1)LTMFHGR | ......AK...... / ......EK...... / ......KE...... | -2.02 | -5.46 |

**Figure 4.  Protein domains and lysine footprints of UPL3-associated differentially ubiquitin conjugates (|FC| > 1.2, *P*-value < 0.05).**
**(A)** GO terms enrichment analysis of enriched protein domains of differentially ubiquitin conjugates (|FC| > 1.2, *P*-value < 0.05) via InterProScan. **(B)** The consensus ubiquitin attachment motif identified by the MEME Suite. **(C)** List of apparent ubiquitin attachment sites and motif in identified proteins (|FC| > 1.5, *P*-value < 0.05). R1 and R2 are two replicates. Those up-regulated and down-regulated in *upl3* relative to WT were highlighted in red and blue, respectively.

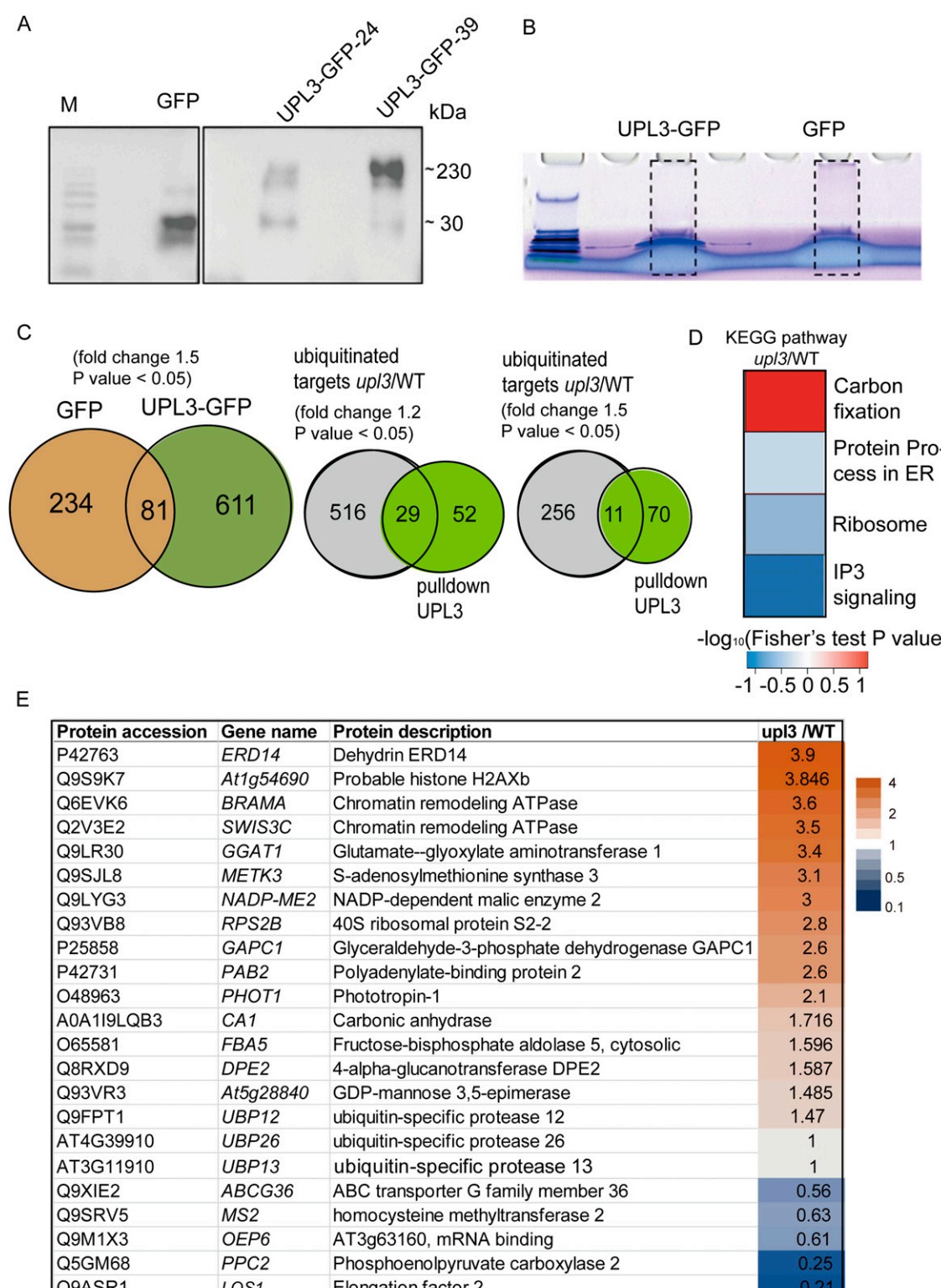

**Figure 5. UPL3-interacting proteins overlapping with differentially ubiquitin conjugates (DUCs) identify potential UPL3-ubiquitinated substrates ABCG36, MS2, OEP6, PPC2, and LOS1.**
**(A)** Immunodetection of protein samples isolated from GFP- and UPL3-GFP–expressing plants with an antibody against GFP. **(B)** Nano-trapped proteins for MS analysis. **(C)** Overlapping proteins between UPL3-interacting candidates and the DUCs. Eighty-one putative interacting proteins were identified as specific to UPL3-GFP, which contained 29 and 11 DUCs with a fold-change |FC| > 1.2 or 1.5, respectively. **(D)** Enrichment of the KEGG pathway in the overlapping genes highlights an involvement of potential UPL3-ubiquitinated substrates in the IP3K pathway and in ribosomal activity. Red color indicates up-regulated, and blue color indicates down-regulated. **(E)** List

HXK1, PPC2, ABCG36, MS2, LOS1 exhibited a down-regulated 1.5–3-fold in the *upl3* mutant relative to WT (Supplemental Data 5). Furthermore, when the ubiquitination enrichment was normalized to the protein level, the level of ubiquitinated UBP12, UBP13, and UBP26 was not significantly changed between the *upl3* and WT; however, the ubiquitin level of ubiquitinated UBC7, UBC35, and BRM maintained an up-regulated potential. HXK1, PPC2, ABCG36, MS2, and LOS1 maintained a down-regulated trend (Fig 7C). This suggests that UBP12 and other UBP members are interacting partners but were not altered their ubiquitin level by UPL3.

To examine whether the interaction between UPL3 and UBP12 might contribute to the protein level of ubiquitinated targets in *planta*, the loss- or gain-of-*UPL3* and -*UBP12* plants were used. We first detected the protein level of BRM, HXK1, and PPC2 in the *ubp12* and the *upl3-1* mutant background and overexpressing *UBP12* and *UPL3-24* transgenic plants relative to WT using antibodies against BRM, HXK1, PPC2, and polyubiquitin. The protein levels of PPC2 and HXK1 were increased by *UPL3* mutation and in overexpressing *UBP12* plant and decreased by *UPL3* overexpression and by *UBP12* mutation (Fig 7C). The total ubiquitination level in the *upl3* or the *ubp12* was comparable or slightly stronger than in the WT, whereas it declined considerably in the *UPL3*- or *UBP12*-overexpressing line (Fig 7C), In contrast, the BRM protein level was increased in both *upl3* and *oeUPL3* but however was down-regulated in the *oeUBP12* and did not change in the *ubp12*. To avoid an influence of UPL3 or UBP12 on *BRM*, *PPC2* and *HKX1* gene expression at the transcriptional level, the transcript levels of *BRM*, *PPC2*, and *HXK1* in loss- or gain-of-*UPL3* and -*UBP12* mutants were detected by RT-qPCR and showed there were no significantly different of PPC2 and HXK1, but the transcript level of *BRM* significantly up-regulated in the *oeUBP12* and down-regulated in the *ubp12* (Fig S10). It suggested that UPL3 affected PPC2 and HXK1 protein levels in the UPS degradation pathway, although UPL3 unexpectedly declined globally polyubiquitination enrichment (Figs 2 and 7C). However, UBP12 and UPL3 displayed the opposite effect on HXK1 and PPC2 protein levels when their genes were mutated or overexpression. Loss of *UBP12* or gain of *UPL3* reduced but loss of *UPL3* or gain of *UBP12* increased the protein accumulation relative to WT (Fig 7C). The BRM level was regulated by UBP12 in transcriptional and protein levels. Furthermore, during treatment with inhibitor MG132 of protein ubiquitination in the *upl3* and the *oeUPL3* plants, the immunodetection of the protein level of PPC2 and HXK1 showed that two protein levels were pronouncedly accumulated in the *upl3* and the *oeUPL3* after treatment, although PPC2 and HXK1 levels were higher in the *upl3* and lower in the *oeUPL3* plants relative to WT (Fig 7D). The BRM level was not affected by MG132 treatment in the *upl3* and *ubp12* background but significantly up-regulated in *oeUBP12* and *oeUPL3* plants after MG132 treatment. Notably, the polyubiquitin level was apparently increased in the *upl3* lines but decreased in the *oeUPL3* lines; however, its level was enhanced in the *upl3* and declined in the *oeUPL3* lines after the treatment of MG132, suggesting that UPL3 has transferring ubiquitin activity, but it did not directly affect the

BRM protein level. We concluded that UPL3 has a major effect on PPC2 and HXK1 protein degradation, whereas UBP12 antagonistically affect the HXK1 and PPC2 protein levels and BRM level in the transcriptional level and protein level. These results suggest a complicated molecular interaction network between UPL3 and these proteins, probably in terms of homeostasis in protein ubiquitination.

Because PPC2 is a key enzyme for primary metabolism, the carbohydrate contents of the *upl3* and *ubp12* mutant plants was monitored to check whether the opposite phenotypes observed in *upl3* and *ubp12* were related to carbon metabolism in the *ppc2* and the *gin2*, a *HXK1* mutant *gin2* (*glucose-insensitive2*) (Cho et al, 2006). The *upl3* mutant showed a decreased starch and sucrose accumulation compared with WT, whereas complementation or overexpression of *UPL3* restored the starch and sucrose content to the WT level (Fig 7E–H). In contrast, the *ubp12* mutant displayed significantly stronger starch and sucrose accumulation than that in WT, whereas effect of *UBP12* overexpression was similar to that of the *upl3* mutant, with a much lower starch and sucrose content in the plants (Fig 7E–H). The contents of starch and sucrose were significant accumulated in the *ppc2* and the *gin2* mutant compared with WT. Therefore, starch accumulation is positively correlated with mutations in the two antagonistic ubiquitination/deubiquitination pathway genes.

## The UPL3 interacting UBP12 regulates carbohydrate metabolism related leaf aging and flowering

We further sought insights from phenotypic analysis of the *upl3-1*, *brm*, *ubp12*, *gin2-1*, and *ppc2* mutant plants. The results showed that *brm*, *ubp12*, *gin2*, and *ppc2* plants shared a common phenotype of lower numbers of rosette leaves and early bolting and flowering, but *BRM* and *PPC2* caused the curled leaves and premature leaf aging phenotype only found in the *ppc2* and the *brm* plants (Fig 8A and B). Summarily, *upl3* plants displayed a late-senescence phenotype with more numbers of rosette leaves and delayed flowering time (more than 1 wk); the effect of UPL3 on plant development contrasted those of both PPC2 and HXK1 (Figs 8C and D and S11), thereby a mild curled-leaves phenotype was also observed in aging leaves of the *upl3* plants at a later stage of development (Figs 1A and 8A), consistent with previous reports of phenotype of *brm*, *ubp12*, and *ppc2* mutants (Cui et al, 2013; Xu et al, 2016; Li et al, 2016a, 2016b; Archacki et al, 2017; Park et al, 2019).

Based on a delaying senescence phenotype of *upl3* mutant and premature senescence and early bolting phenotype of *ubp12* mutant, we further selected 13 genes coding for transcription factors such as WRKYs and NACs related to leaf aging (e.g., WRKY53, WRKY75, and ORE1) and salicylic acid responsive senescence (e.g., WRKY38, WRKY63, and WRKY51 etc.), as well as floral transition genes (e.g., FT and FLC) for RT-qPCR analysis (Fig 8G and H). The results showed that the expression level of *WRKY53*, *WRKY75*, *WRKY38*, *WRKY63*, and *WRKY51*, as well as *NACs* was significantly decreased

of most notable interacting candidates of UPL3 with their respective fold-change of ubiquitination level in *upl3* versus wild-type plants. Orange indicates up, and blue color indicates down. The fold-change is indicated with the data.
Source data are available for this figure.

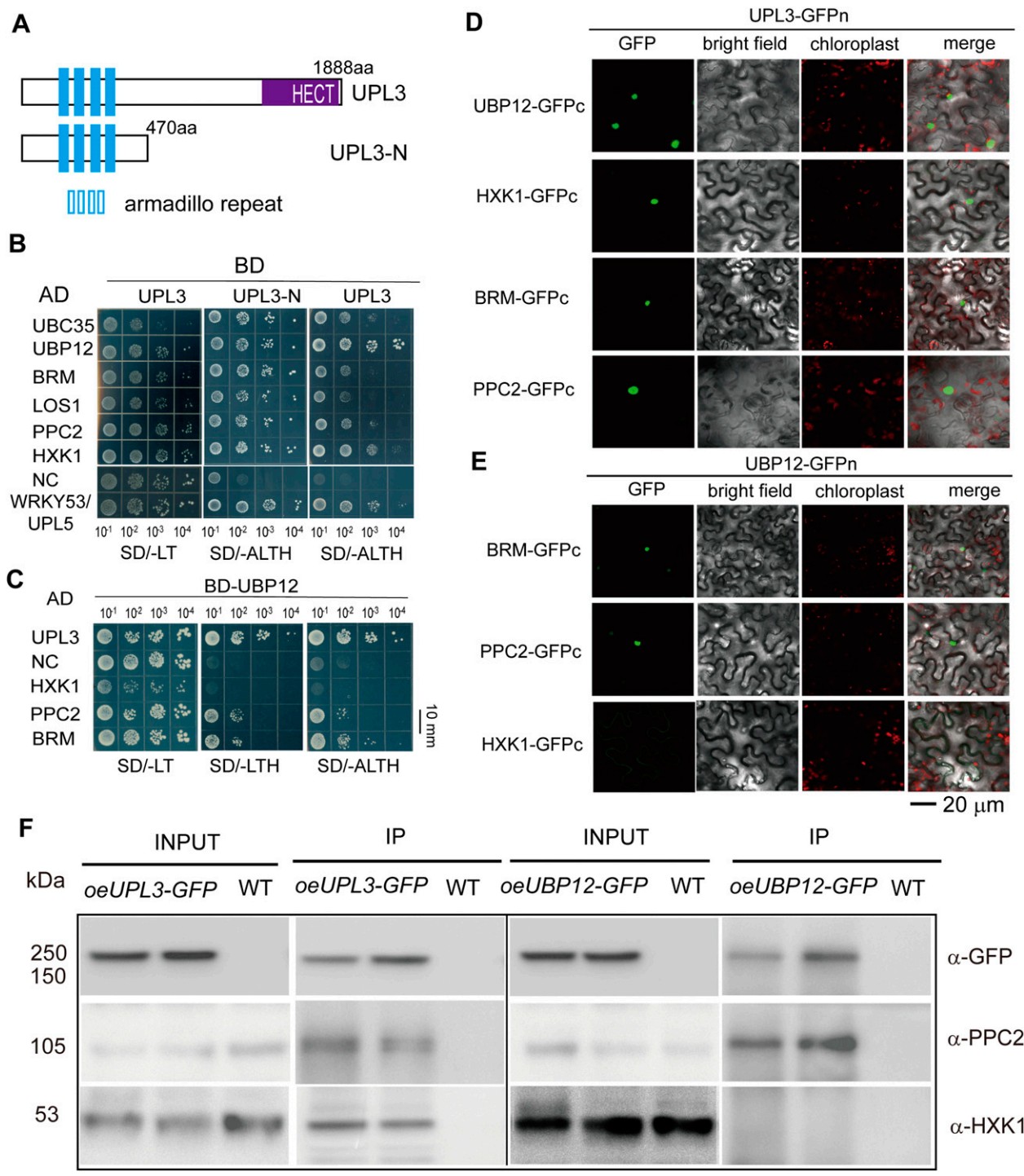

**Figure 6. Confirmation and characterization of UPL3 interaction with its targets.**
**(A)** Domain structure of the UPL3 protein showing the full-length and the N-terminus used in Y2H assay. **(B)** Confirmation of UPL3 interaction with its targets in a yeast two-hybrid assay. The AD or BD empty vector was used as a negative control, and the interacting pair UPL5-WRKY53 was used as a positive control. SD/-LT: SD media minus Leu and Trp, SD/-ALTH: SD media minus Ade, Leu, Trp, and His. Bars = 10 mm. **(C)** UBP12 interacts with BRM, PPC2, but not HXK1 in a yeast two-hybrid assay. SD/-LT: SD media minus Leu and Trp, SD/-LTH: SD media minus Leu, Trp, and His, SD/-ALTH: SD media minus Ade, Leu, Trp, and His. Bars = 10 mm. **(D, E)** The detection of bimolecular fluorescence complementary assay; an empty vector was used as a negative control (Fig S8B). Bars = 20 μm. **(F)** CoIP detection of ubiquitin conjugates (representation: HXK1 and PPC2) in the overexpression UPL3-GFP and UBP12-GFP plants compared with wild-type plants. Antibodies against HXK1 and PPC2 (Agrisera) and anti-GFP (Rothe) are used.
Source data are available for this figure.

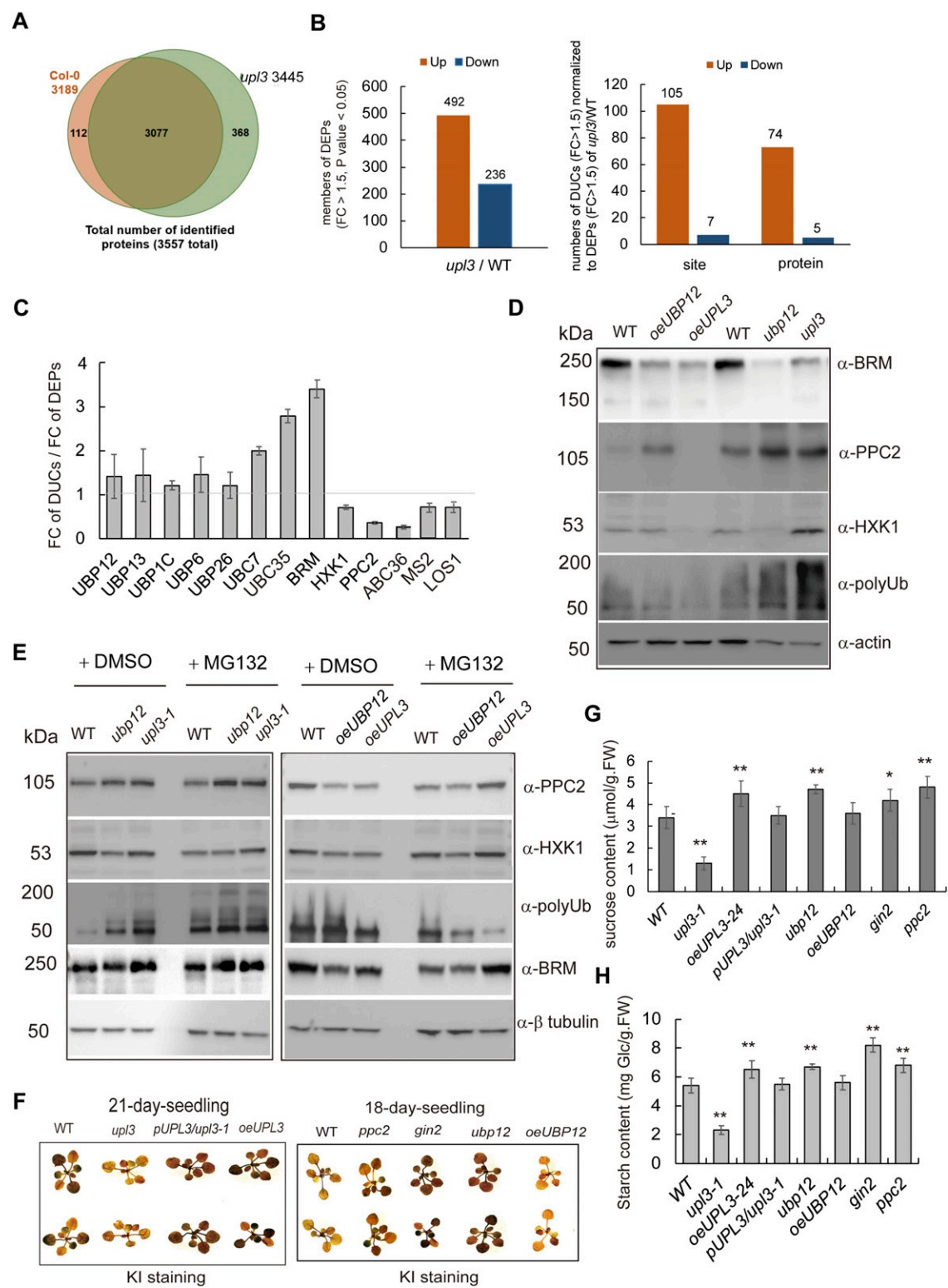

**Figure 7. The protein levels of representation: BRM, HXK1, and PPC2 in the *UPL3* or *UBP12* mutants.**
**(A)** A total of 3,557 Arabidopsis proteins could be reproducibly identified and quantified in both samples by our liquid chromatography–mass spectrometry (LC–MS) regime analyzed in total triplicate, 3,445 of which were described here as targets of UPL3-associated DEPs (cutoff, *P*-value < 0.05). **(B)** Numbers of differentially expression proteins (DEPs) (|FC| > 1.5, *P*-value < 0.05) of *upl3* relative to WT (Left). Numbers of differentially ubiquitinated conjugates (DUCs) (|FC| > 1.2, *P*-value < 0.05) normalized to DEPs (|FC| > 1.5, *P*-value < 0.05) (Right). Orange, up-regulated; blue, down-regulated. **(C)** The fold-change of UBP proteins and BRM, PPC1, PPC2, UBC2, UBC7, HXK1 from proteome dataset of *upl3*/WT (Supplemental Data 5). **(D)** Immuno-detection of ubiquitin conjugates (representation: BRM, HXK1, and PPC2) in the *upl3* and the *ubp12*; overexpression

either in the *upl3* (Fig 8G and H) or in the *oeUBP12*, in contrast significantly increased either in the *oeUPL3* or in the *ubp12*. These data confirmed the notion that UPL3 had a profound functional involvement in stress-responsive cell senescence and developmental cell senescence (aging) via ubiquitination of their regulators either directly or indirectly.

Because UPL3 interacts with its partners in the nucleus, we compared transcriptome data of the *upl3* and WT plants, which released from Furniss et al (2018) (Supplemental Data 6). A total of 1,467 differentially expressed genes between the *upl3* and WT seedlings were identified (Figs S12A and Supplemental Data 6). Of these, genes related to stress responses (drug, hypoxia, oxidative, and UV) were most significantly up-regulated, followed by genes associated with cellular glucan metabolic processes, anthocyanin-containing compound biosynthesis, and cellular polysaccharide catabolic processes. A third set of up-regulated genes was involved in protein transport and leaf senescence. In contrast, genes for response to (a)biotic stress (metal ion salt stress and fungus) and plant aging were most significantly down-regulated, followed by genes related to secondary metabolic processes, MAPK signaling pathway, leaf cell death, and phenylpropanoid biosynthesis (Figs 8H and S12B).

We analyzed DUCs and DEGs datasets of *upl3*/WT and showed that 26 proteins were identified as the overlapping proteins of DEGs and DUCs, with an altered ubiquitinated protein level (|FC| > 1.5, *P*-value < 0.05) and gene expression level (|FC| > 1.3, *P*-value < 0.05) by the *upl3* mutation (Fig S12A). Among these proteins, four proteins (Fig S12B, fold-change depicted with two orange values) with increased ubiquitin conjugates were up-regulated in gene expression in the *upl3* background, including glucomannan 4-*β*-mannosyl-transferase 9 (CSLA9), inositol-3-phosphate synthase isozyme 1 (MIPS1), calcium-binding protein 16 (CML16), and the Patellin-2 (PATL2) (Fig S12B, fold-change depicted with two orange values). Two other proteins (Fig S12B, fold-change depicted with a blue and an orange value) with reduced ubiquitin conjugates but with an increased transcript level in the *upl3* mutant were cysteine lyase (CORI3) and cinnamyl alcohol dehydrogenase 7 (CAD7), both of which had reported functions in the amino acid metabolic process (Tsuwamoto & Harada, 2011; Tanaka et al, 2018). On the other hand, 9 proteins (Fig S12B, fold-change depicted with an orange and a blue value) showed an enhanced ubiquitin conjugates level but a down-regulated expression level in the *upl3* mutant plants. These included calmodulin-like protein10 (CML10), aquaporin (PIP1-5), triacylglycerol lipase-like 1 (TLL1), leucine-rich repeat ser/thr protein kinase (LRR-RLK), the jacalin-related lectin 23 (JAL23), ABCB transporter member 19 (ABCB19), methionine aminotransferase (BCAT4), and the ankyrin repeat-containing protein (BDA1). The remaining 11 proteins (Fig S12B, fold-change depicted in blue) were those with both a reduced ubiquitin conjugates level and down-regulated gene expression in the *upl3* mutant. Their

functions seemed to be related to glycoside metabolism and transport pathways. From these data, it is obvious that the UPL3-centered molecular network involves both feed-forward and feed-back regulatory pathways and mostly impacts on cellular metabolism, stress-responsive cell death, and aging.

# Discussion

During cell senescence, the predominant regulation is protein degradation, which is critical not only in signal transduction but also for the execution of the senescence syndrome (Buchanan-Wollaston et al, 2005; Woodson et al, 2015; Yolcu et al, 2017; Guo et al, 2021). Our multi-omics analysis provides an overview of UPL3 downstream targets that were either directly or indirectly affected at the protein level and the transcript level. An unexpected finding is that loss of UPL3 results in a globally enhanced ubiquitination of metabolism-related proteins. The unveiling of physical interactions with the ubiquitin-specific proteases UBP12, BRM, HXK1, and PPC2 is of particular interest, suggesting a potential action model of UPL3 interacting with UBP12 in BRM, HXK1, and PPC2, mediating carbohydrate metabolic–related protein turnover during plant development (Fig 9).

Globally, the loss of the HECT-type E3 ubiquitin ligase *UPL3* leads to the depletion of ubiquitin conjugates of specific proteins, accounting for ~1/3 of the total proteins whose ubiquitination status is significantly altered, with a ratio |FC| > 1.2 (Fig 2). And the proteome of the *upl3*, accounting for 1,737 proteins, is significantly altered with a fold-change |FC| > 1.2 (Supplemental Data 5). Although two combined proteomic and ubiquitomic datasets of three replicates are used to analyze and the variant *P*-value is not so low about 0.6, we harvested more 500 DUCs (|FC| > 1.2, *P*-value < 0.05). After normalized to proteome dataset, these proteins are considered as putative candidates regulated by the UPL3 enzyme, although not all proven experimentally yet. And some of these are demonstrated UPL3 targets, for example, GLS3 (Saracco et al, 2009; Patra et al, 2013; Kim et al, 2015; Gao et al, 2017). It did not appear in our dataset; however, other common known ubiquitinated targets included in the list are AHA1, CDC48A, ERD4, LEC2, and RPN10 proteins (Fig S4, and Ref. in Downes et al [2003], Park et al [2008], Finley [2009], Rai et al [2012], Besche et al [2014], Yamauchi et al [2016], Furniss et al [2018], and Kumari et al [2019]), playing roles in cell division, cell senescence, and stress-induced cell senescence.

### UPL3 alters the levels of ubiquitin conjugates involved in metabolism and heavy-metal stress-induced cell senescence

UPL3 has several new putative targets, such as MS2, OASA1, ALDHA3, HXK1, and PPC2. The protein level of these five putative target

---

*UPL3* and *UBP12* plants compared with wild-type plants. Antibodies against BRM (provided by Dr. Rongcheng Lin), PPC2, and HXK1 (Agrisera) and polyubiquitin (Cell Signaling Tech) are used, and the tubulin level is used as protein-loading controls. **(E)** Immuno-detection of ubiquitin conjugates (representation: BRM, HXK1, and PPC2) in the *upl3* and the *ubp12*; overexpression *UPL3* and *UBP12* plants compared with wild-type plants after treatment of ubiquitination inhibitor MG132. **(F, G, H)** Starch staining with KI and quantification of starch and sucrose in 18–21-d-old rosettes (n = 5). Error bar represents the SD of triplicates. Asterisks denote the statistically significant level different with the wild type, as verified via *t* test: *$P$, 0.05; **$P$, 0.01, n = 5.
Source data are available for this figure.

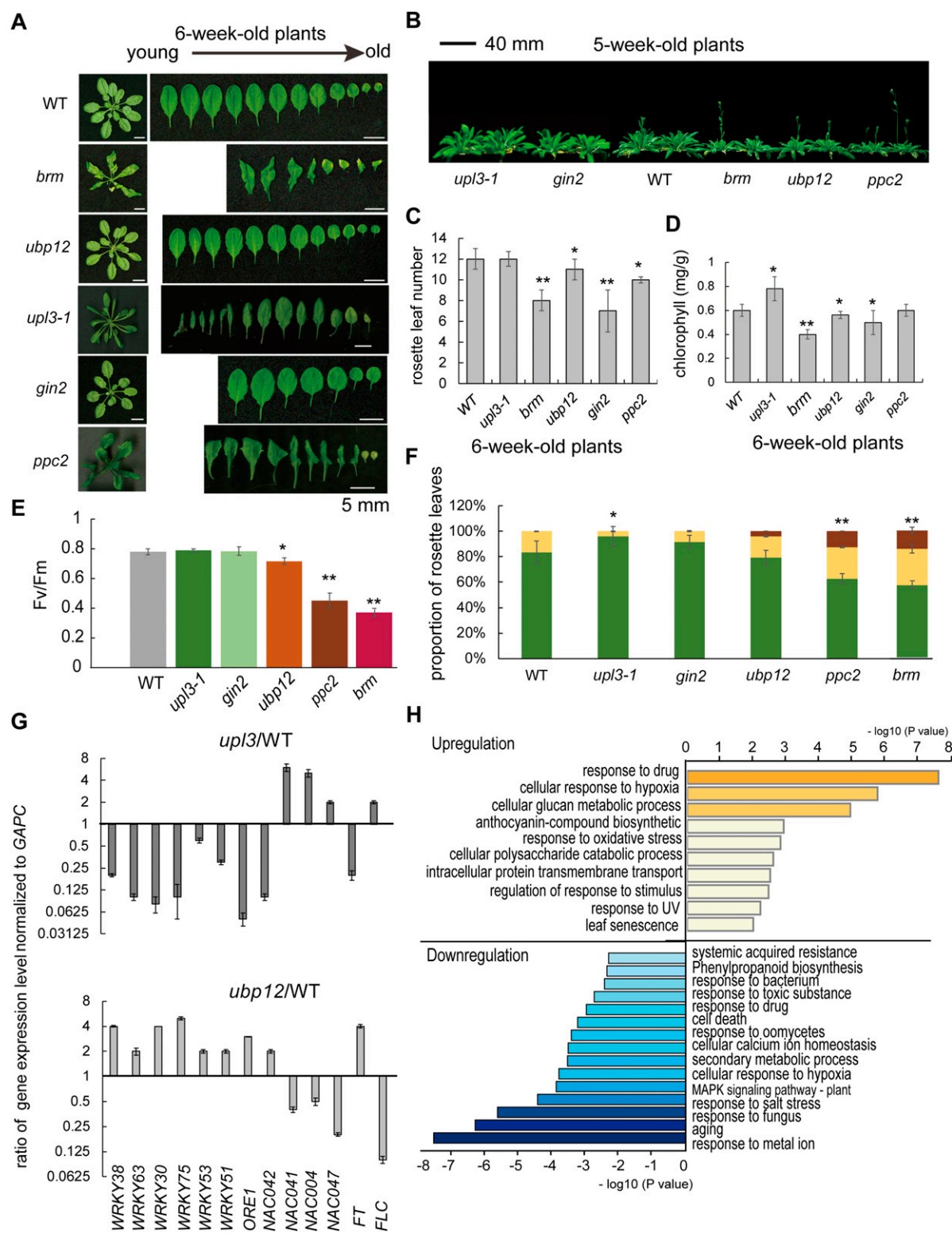

**Figure 8. The *brm*, *ppc2*, *ubp12*, and *oeUPL3* plants show similar early leaf senescence and early-flowering phenotypes.**
**(A)** Phenotypes of 5-wk-old plants. Bars = 5 mm. **(B)** Bolting and flowering phenotype of the 5-wk-old plants. Bars = 40 mm. **(C)** Rosette leaf numbers of different genotypes at indicated time (n > 20). **(D)** Chlorophyll content in rosette leaves of 6-wk-old plants (n = 3). **(E)** Photosystem II fluorescence activity (Fv/Fm) of rosette leaves of 6-wk-old plants (n = 5). **(F)** Proportion of green and yellow leaves of the whole rosette of 8-wk-old plants (n = 12). **(G)** Quantitative real-time PCR of gene expression from 6-wk-old *upl3* and wild-type plants for 13 selected plant aging and senescence-related transcription factors. Error bars represent the SD of three biological replicates. **(H)** GO terms enrichment analysis of 1,467 differential display genes (DEGs, |FC| > 2, *P*-value < 0.05) of *upl3*/WT.

proteins were accumulated by proteome analysis (Fig 7C and Supplemental Data 5), and the ubiquitin level of them were down-regulated in the *upl3* mutant relative to WT by ubiquitome analysis (Figs 2 and 3 and Supplemental Data 2); the ubiquitinated sites were identified as a noncanonical pattern (Fig 4); two of them: HXK1 and PPC2 were detected to be direct interacting partners of UPL3 (Fig 6). This evidence supposed that UPL3 recruited them and affects their ubiquitin level and protein level in the ubiquitination/26S proteasome system (UPS) degradation pathway. It has been reported that these five putative proteins play important roles in promoting cell proliferation and expansion during early leaf development (HXK1, Van Dingenen et al, 2019), delaying flowering and leaf senescence (ALDHA3, Stiti et al, 2011), reducing starch accumulation (PPC2, Shi et al, 2015; You et al, 2020), and lowering cadmium tolerance, defense response against abiotic stresses such as salicylic acid, salinity, heavy metal, and leaf cell death (OASA1, Dominguez-Solís et al, 2001; Shirzadian-Khorramabad et al, 2010; Birke et al, 2013), which is consistent with the *upl3* phenotype (Figs 1 and 8; Furniss et al, 2018). Therefore, HXK1 and PPC2 might be direct targets of UPL3 to affect metabolite flux in carbon fixation in C3 plant such as *Arabidopsis*. In most nonphotosynthetic tissues and the photosynthetic tissues of C3 plants, the fundamental function of PPC2 is to anaplerotically replenish tricarboxylic acid cycle intermediates, which can be activated by its positive effector, Glc-6-P, and inhibited by its negative effectors, such as malate, Asp, and Glu, as well as the levels of glycine and serine (O'Leary et al, 2011; You et al, 2020). Our result showed that UPL3 also affected the ubiquitin level of MS2 and OASA1 in methionine and cysteine biosynthesis pathways (Fig 3), which feedbacks perhaps negatively regulates the PPC2 protein level through carbohydrate and amino acid biosynthetic process.

### UBP12 is an UPL3-interacting protein, which in-turn might act as an important downstream player in protein ubiquitination

As described in previous review, the HECT E3 N-terminus not only simply serves as an adapter for direct binding but it also provides various ways to regulate HECT E3s' substrate recruitment and catalytic activity (Lan & Miao, 2019; Wang & Spoel, 2022). In this study, we showed that UPL3 seemed to interact with histone variants, histone modifiers (BRM), PPC2, HXK1, several U-box proteins (UBC35), and DUBs (UBP12) in the nucleus (Figs 6 and S8). The 2/3 ubiquitin conjugates (DUCs) including BRM, histone variants, and UBC35 were all enhanced in the *upl3* plants (Fig 2). If we only look at canonical function of UPL3 E3 ligase in the UPS pathway, it seems to be hard to explain the UPL3 affecting the polyubiquitination pattern. Furniss et al (2018) suggested that UPL3 might have E4 function, recruiting polyubiquitin to targets (Furniss et al, 2018); however, it could not explain our issue yet. This study showed that UPL3 could interact with UBP12, UBP13, and UBP26 (Figs 5 and 6); however, the ubiquitin level of UBP12, UBP13, UBP26, and other members were not significantly altered in the *upl3* (Fig 7). Thus, we supposed that UBP12, UBP13, and UBP26 might be recruited by UPL3 for nuclear access or playing their roles together. In fact, it has reported that UBP12 has function in the protein deubiquitination and is able to interact in vivo with a Polycomb G protein (LHP1) and EMBRYONIC FLOWER1 complex (EMF1c) to form a complex and make

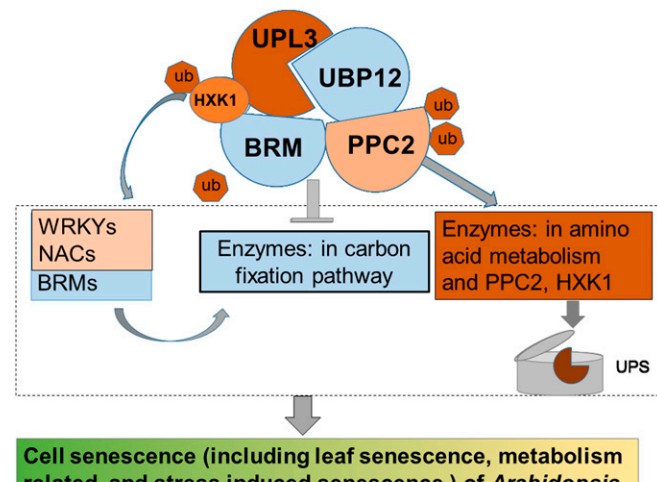

**Figure 9. A noncanonical working model of UPL3 in regulating cell senescence of *Arabidopsis* by multi-omics and genetics analysis.**
Based on integrative datasets of ubiquitome, proteome, and transcriptome. Preferential ubiquitination of proteins related to carbon fixation represented the largest set of proteins with increased ubiquitination in the *upl3* plant, whereas a small set of proteins with reduced ubiquitination caused by the *upl3* mutation were linked to cysteine/methionine synthesis processes. Notably, ubiquitin hydrolase 12 (UBP12), BRM, HXK1, and PPC2 were among the UPL3-interacting partners identified as the UPL3-interacting partners by both GFP nanotrap–mass spectrometry analyses and yeast two-hybrid assay; characterization of *upl3*-, *brm*-, *ppc2*-, *gin2*-, and *ubp12*-mutant plants and transcriptome analysis suggested that UPL3 fine-tunes carbon metabolism, mediating cellular senescence via proteolysis-independent regulation and proteolysis-dependent degradation on metabolism-mediated cell senescence. Red color represents up-regulated, blue color represents down-regulated, thick frame represents more numbers of ubiquitinated proteins.

removal of H2A ubiquitin, being necessary for transcriptional regulation such as BRM, several WRKYs, and NACs transcription factors in plant senescence and flowering and repress genes involved in stimulus response (Figs 8 and S10) (Li et al, 2016a; Miao & Zentgraf, 2007; Guo et al, 2017; Zhang et al, 2017; Kralemann et al, 2020). In this study, similar phenotypes among *oeUPL3*, *brm*, *ubp12*, *ppc2*, and *gin2* mutants or between *upl3* and *oeUBP12* are helpful to explain UPL3 and UBP12 playing their roles together to determine ubiquitination status of histone variants or candidate targets for correct biological function in plant senescence and flowering and organ development. *oeUPL3* had more similar phenotype with the *ubp12 ubp13* double-mutant and *brm* mutant: early flowering (Cui et al, 2013; An et al, 2018), premature leaf aging phenotype (Park et al, 2019; Vanhaeren et al, 2020), and response to SA (Furniss et al, 2018) and JA (Ewan et al, 2011; Jeong et al, 2017), as well as plant defense response (Ewan et al, 2011), starch and sucrose accumulation (Figs 7 and 8), exhibiting a delayed senescence, and flowering phenotype (Cao et al, 2008; Kotliński et al, 2017; Rutowicz et al, 2019; Kralemann et al, 2020), but the transcript level of *BRM*, *HXK1*, and *PPC2* was not regulated by UPL3, although a series of transcription factors, such as WRKYs, ORE1, and NACs, as well as CML10, PIP1-5, TLL1, LRR-RLK, JAL23, ABCB19, BCAT4, and BDA1, whose functions were involved in response to (a)biotic stress and in plant development and plant aging in response to light, auxin, JA, or calcium (Wu et al, 2010; Yang et al, 2012, 2013; Debernardi et al, 2014),

are parts of common downstream genes of UPL3 and UBP12/13 (Figs 8 and S12; Lee et al, 2019). The functions of UPL3 interacting with UBP12 might have two possibilities: (1) recruiting UBP12 in enhancing the deubiquitination of a large set of proteins, such as carbon metabolism–related enzymes; (2) recruiting UBP12 in chromatin accessing and activating downstream gene expression. The detailed insight into the regulatory mechanism of ubiquitination dynamic between UPL3 and UBP12 with their substrate complex would be the next challenge in the field.

### The UPL3 active site is dependent on the noncanonical ubiquitination motif

Based on the analysis of the Kub site information in this study, the canonical c-K-x-E/G motif sites are highly enriched in the ubiquitin conjugates in the upl3, whereas the noncanonical motif −x-A-K-x-sites show low enrichment for HXK1 and PPC2 (Fig 4), because HXK1 at K77 was previously reported to be ubiquitinated at a site of noncanonical linkages in animal cell (Huang et al, 2018). Thus, our omics results confirm the assembly of polyubiquitin chains in plants. By scanning our ubiquitome datasets for ubiquitination sites using footprints containing ubiquitin remnants after trypsin cleavage, modifications by SUMO1 at K23 and K42 were detected in addition to polyubiquitin linked via K48 linkages. Furthermore, according to acetylation of lysine (Kac) sites of GAPDH protein both in human and in rice under stress condition is required for protein translocation into the nucleus (Boukouris et al, 2016; Li et al, 2016a, 2016b; Huang et al, 2018). Our data presented here indicate that one lysine (K76 in AtGAPC) was ubiquitinated in the upl3 plants. Perhaps, it is competed this lysine site for acetylation or phosphorylation (Swaney et al, 2013; Elia et al, 2015; Ohtake et al, 2015). However, different extracellular stimuli and their intracellular signaling messengers (e.g., heat, salt, osmotic stress, heavy metal, reactive oxygen species, and lipid mediators) might differentially and specifically induce ubiquitination of GAPC for nuclear access via the Kub site pattern (Li et al, 2016a, 2016b; Kim et al, 2020). From this study, several proteins such as BRM, BCAT4, and SAM1 and different lysine sites showed various signs of up- or down-regulated ubiquitination (Fig 2E). Therefore, we speculate that the Kub site preference pattern of UPL3 mediates their function and regulation. It can be explained that 80% ubiquitinated proteins are enriched outside of nucleus.

# Materials and Methods

### Materials

The seeds of Arabidopsis (A. thaliana) Col-0 and transgenic plants were germinated, and seedlings were grown on half-strength Murashige–Skoog (MS) medium supplemented with 0.7% (wt/vol) agar. Ten-d-old seedlings were transplanted to the vermiculite, watering with half-strength MS medium in climate rooms under controlled conditions (22.5°C, 13/11 h of light/dark photoperiod with a light intensity of 80 $\mu$mol photons m$^{-2}$ s$^{-1}$, 60% relative humidity). The mutant seeds of upl3-1 (SALK_015334) and upl3-3

(SALK_117247) were obtained from the Nottingham Arabidopsis Stock Centre (NASC).

UPL3 or UBP12 overexpression (oeUPL3) and complementation plants (comUPL3, pUPL3/upl3) were constructed by cloning the UPL3 or UBP12 coding sequence into pCAMBIA3301 vector using SpeI and SmaI under ACTIN3 promoter and its own promoter P$_{UPL3}$, which contains 2 kb upstream of the ATG start codon) with the primers described in Table S1. Arabidopsis transformation was performed by the floral-dip method (Clough & Bent, 1998). The homozygous upl3 T-DNA insertion knockout mutants were obtained by genome insertion screening and RNA-level screening (Fig S1) with the primers described in Table S2. The brm, gin2, ppc2, ubp12, ubp12 ubp13, and oeUBP12 seeds were kindly provided by other scientists.

### Starch and glucose measurement and staining

Whole rosettes of 18–21-d ubp12 and upl3 transgenic plants were either harvested or covered with black trays at 10 AM after 2 h. At 10 AM of the next day, rosettes of covered plants were harvested. Rosettes were cleared in 80% (vol/vol) ethanol plus 5% (vol/vol) formic acid at room temperature, stained in I$_2$-KI solution (5 g I$_2$ and 10 g KI per 100 ml sterile water), and washed three times in water (Huang, et al, 2020). The contents of starch and sucrose were measured according to the method described by Huang et al (2020)

### Protein extraction, trypsin digestion, HPLC fractionation, affinity enrichment, and LC–MS/MS analysis

The rosette leaves of 10 6-wk-old upl3 and wild-type plants were pooled and ground in liquid nitrogen into cell powder and then transferred to a 5-ml centrifuge tube, collected in three tubes for one biological replicates; a total of 30 plants were collected for three biological replicates. After that, the method described according to the experimental manual of proteomic/ubiquitinomics determination in the "Jingjie Bio" company was followed.

Four volumes of lysis buffer (8 M urea, 1% Triton-100, 10 mM dithiothreitol, and 1% Protease Inhibitor Cocktail [Roche]) was added to the cell powder, followed by sonication three times on ice using a high-intensity ultrasonic processor (Scientz). For PTM experiments, inhibitors were also added to the lysis buffer, for example, 3 $\mu$M TSA and 50 mM NAM for acetylation. The remaining debris was removed by centrifugation at 20,000$g$ at 4°C for 10 min. Finally, the protein was precipitated with cold 20% TCA for 2 h at −20°C. After centrifugation at 12,000$g$ 4°C for 10 min, the supernatant was discarded. The remaining precipitate was washed with cold acetone for three times. The protein was redissolved in 8 M urea, and the protein concentration was determined with a BCA kit according to the manufacturer's instructions.

### Trypsin digestion

A total of 5 mM dithiothreitol was added to the protein solution for reduction for 30 min at 56°C, and the solution was alkylated with 11 mM iodoacetamide for 15 min at room temperature in darkness. The protein sample was then diluted by adding 100 mM NH4HCO3 to urea concentration less than 2 M. Finally, trypsin was added at 1:50 trypsin-to-protein-mass ratio for the first digestion overnight and

1:100 trypsin-to-protein-mass ratio for a second 4-h digestion. Approximately 100 mg of protein for each sample was digested with trypsin for the following experiments.

## HPLC fractionation

The tryptic peptides were fractionated by high-pH reverse-phase HPLC using the Thermo BetaSil C18 column (5-$\mu$m particles, 10 mm ID, 250 mm length). Briefly, peptides were first separated with a gradient of 8–32% acetonitrile (pH 9.0) over 60 min into 60 fractions. Then, the peptides were combined into 18 fractions and dried by vacuum centrifugation.

## Affinity enrichment

To enrich Kub peptides, tryptic peptides dissolved in NETN buffer (100 mM NaCl, 1 mM EDTA, 50 mM Tris–HCl, and 0.5% Nonidet P-40, pH 8) were incubated with prewashed antibody beads at 4°C overnight with gentle shaking; the di-Gly-Lys specific antibody was used (PTM Biolabs). Then the beads were washed four times with NETN buffer and twice with distilled, deionized water. The bound peptides were eluted from the beads with 0.1% trifluoroacetic acid. The eluted fractions were combined and vacuum-dried. The resulting peptides were cleaned with C18 ZipTips (Millipore) according to the manufacturer's instructions, followed by LC–MS/MS analysis.

## LC–MS/MS analysis

Three parallel analyses for each fraction were performed. LC–MS/MS analysis was performed according to previously described protocols (Xie et al, 2015). The tryptic peptides were dissolved in 0.1% formic acid (solvent A), directly loaded onto a home-made reverse-phase analytical column (15-cm length, 75 $\mu$m i.d.). The gradient was from 6% to 23% solvent B (0.1% formic acid in 98% acetonitrile) over 26 min, 23–35% in 8 min, and climbing to 80% in 3 min and then holding at 80% for the last 3 min, all at a constant flow rate of 400 nl/min on an EASY-nLC 1000 UPLC system.

The peptides were subjected to NSI source followed by tandem mass spectrometry (MS/MS) in Q Exactive TM Plus (Thermo Fisher Scientific) coupled online to the UPLC. The electrospray voltage applied was 2.0 kV. The m/z scan range was 350–1,800 for full scan, and intact peptides were detected in the Orbitrap at a resolution of 70,000. Peptides were then selected for MS/MS using NCE setting of 28, and the fragments were detected in the Orbitrap at a resolution of 17,500.

A data-dependent procedure that alternated between one mass spectrometry scan followed by 20 MS/MS scans was applied for the top 20 precursor ions above a threshold ion count of $1.5 \times 10^4$ in the mass spectrometry survey scan with 30-s dynamic exclusion. The electrospray voltage applied was 2 kV. Automatic gain control was used to prevent overfilling of the ion trap; $5 \times 10^4$ ions were accumulated for the generation of MS/MS spectra. For mass spectrometry scans, the mass-to-charge ratio scan range was 350 to 1,800. The fixed first mass was set as 100 mass-to-charge ratios.

## Database search

The resulting MS/MS data were processed using the Maxquant search engine (v.1.5.2.8) (https://www.maxquant.org). Tandem mass spectra were searched against the Uniprot database concatenated with a reverse-decoy database. Trypsin/P was specified as cleavage enzyme allowing up to four missing cleavages. The mass tolerance for precursor ions was set as 20 ppm in First search and 5 ppm in Main search, and the mass tolerance for fragment ions was set as 0.02 D. Carbamidomethyl on Cys was specified as fixed modification, and GlyGly on Lys and oxidation on Met were specified as variable modifications. The label-free quantification method (Xu et al, 2010) was employed to calculate the relative abundance of the modified peptides. FDR was adjusted to < 1%, and the minimum score for modified peptides was set to >40.

## Ubiquitin footprints

Ubiquitin footprints were identified through Proteome Discoverer (version 2.0.0.802; Thermo Fisher Scientific) by searching the TAIR10 protein database using the variable modification of lysine residues by ubiquitin (Gly–Gly, +114.043 m/z). Peptides were assigned using SEQUEST HT (Thermo Fisher Scientific), with search parameters set to assume trypsin digestion with a maximum of two missed cleavages, a minimum peptide length of 6, precursor mass tolerances of 10 ppm, and fragment mass tolerances of 0.02 D. Carbamidomethylation of cysteine was specified as a static modification, whereas oxidation of methionine and N-terminal acetylation were specified as dynamic modifications. The target FDR of #1% (strict) was used as validation for PSMs and peptides. Proteins that contained similar peptides and which could not be differentiated based on the MS/MS analysis alone were grouped to satisfy the principles of parsimony. MEME Suite 4.11.4 was used to identify the ubiquitin-binding cKxE/D/G motif, whereas the prevalence of these sites was predicted by referring to GPS-ubiquitin (Zhao et al, 2014).

## GFP nanotrap–MS analysis

The rosette leaves of 10 6-wk-old *ACTIN3:UPL3-GFP* overexpression plants and *ACTIN3:GFP* plants were harvested and ground in liquid nitrogen and homogenized in immunoprecipitation (IP) buffer (50 mM Tris–HCl, pH7.4, 150 mM NaCl, 1 mM EDTA, 1% Nonidet P-40, 1 mM PMSF, 10% glycerol, and 1× protease inhibitor cocktail [Roche]). After centrifugation at 16,000*g* for 10 min at 4°C, the supernatant of each sample was mixed with 30 $\mu$l GBP beads (Fig S6) (Rothbauer et al, 2008) and rotated at 4°C for 2 h. GBP beads were pelleted and washed three times with IP buffer. The immunoprecipitated proteins were eluted from the beads with 2× SDS–PAGE sample buffer by heating at 95°C for 10 min. Triplicate protein samples were separated by 15% SDS–PAGE and then extracted for mass spectrometry analysis according to Deng et al (2016). The antibody against GFP was purchased from Roche company.

## Semi-RT-PCR and RT-qPCR

Semi-RT-PCR and RT-qPCR analyses employed the oligonucleotide primers described in Tables S2 and S3. RNA was extracted from

rosette leaves of 6-wk-old *upl3* and wild-type plants, and the following procedure was performed according to the description (Huang et al, 2020). Semi-qRT-PCR was performed on an ARKTIK thermal cycler (Thermo Fisher Scientific). The *GAPC2* (AT1G13440) was chosen as an internal control (25 cycles). PCR products were run on a 1.0% TAE agarose gel. The transcript abundance of RT-qPCR was normalized to that generated with *GAPC2* based on the comparative threshold method (Pfaffl, 2001). Three independent biological replicates with three technical repeats were performed.

### Yeast two-hybrid assay

Yeast two-hybrid assays were performed as described in the manual for the GAL4-based two-hybrid system 3 protocol (Clontech). Full-length or different regions of candidates were cloned into pGADT7-AD vectors to construct prey constructs, and the Bait vector pGBKT7-BD expressed the wild-type UPL3 or UPL3 variants fused to the GAL4 DNA-binding domain (BD). The procedure was according to that described by Huang et al (2020). The primers were listed in Table S4.

### Coimmunoprecipitation and Western blot analyses

The *oeUPL3-GFP*, *oeUBP12-GFP*, and *oeGFP* plants from the fifth to eighth rosette leaves of 6-wk-old plants were harvested and ground in liquid nitrogen. Total proteins were extracted from 1 g of materials and performed CoIP according to the description by Huang et al (2022).

To extract soluble proteins from plant tissue of *oeUPL3*, *upl3*, *ubp12*, and *oeUBP12* and WT plants, 200 mg of leaf material were batch-frozen in liquid nitrogen, ground into powder, resuspended in 100 $\mu$l of extraction buffer (100 mM Tris, pH 7.2, 10% sucrose, 5 mM MgCl$_2$, 5 mM EGTA, protease inhibitor), and centrifuged at 15,000$g$ for 10 min. The supernatant was used for immunoblotting analysis. Antibodies (Agrisera) against polyubiquitin, HXK1, PPC2, and $\beta$-tubulin (CW0098, KWBIO) were used. Anti-BRM was kindly provided by Dr Rongcheng Lin (Institute of Botany, Chinese Academy of Sciences). The procedure was performed according to Miao and Zentgraf (2010). Proteins were separated on 6% acrylamide gels and transferred to nitrocellulose membranes using standard protocols.

### Bioinformatic analysis

Bioinformatic analysis was performed according to previously described protocols (Xie et al, 2015). The detail procedure described as following:

Enrichment of Gene Ontology analysis: Proteins were classified by GO annotation into three categories: biological process, cellular compartment, and molecular function. For each category, a two-tailed Fisher's exact test was employed to test the enrichment of the differentially expressed protein against all identified proteins. The GO with a corrected *P*-value < 0.05 is considered significant.

Enrichment of pathway analysis: Encyclopedia of Genes and Genomes (KEGG) database was used to identify enriched pathways by a two-tailed Fisher's exact test to test the enrichment of the differentially expressed protein against all identified proteins. The pathway with a corrected *P*-value < 0.05 was considered significant.

These pathways were classified into hierarchical categories according to the KEGG website.

Enrichment of protein domain analysis: For each category proteins, InterPro (a resource that provides functional analysis of protein sequences by classifying them into families and predicting the presence of domains and important sites) database was researched, and a two-tailed Fisher's exact test was employed to test the enrichment of the differentially expressed protein against all identified proteins. Protein domains with a corrected *P*-value < 0.05 were considered significant.

Enrichment-based clustering: For further hierarchical clustering based on differentially expressed protein functional classification (such as: GO, Domain, Pathway, Complex), we first collated all the categories obtained after enrichment along with their *P*-values and then filtered for those categories which were at least enriched in one of the clusters with *P*-value < 0.05. This filtered *P*-value matrix was transformed by the function x = $-\log_{10}$ (*P*-value). Finally, these x values were z-transformed for each functional category. These z scores were then clustered by one-way hierarchical clustering (Euclidean distance, average linkage clustering) in Genesis. Cluster membership was visualized by a heatmap using the "heatmap.2" function from the "gplots" R-package.

### Summary

Bioinformatic analysis was performed according to previously described protocols (Xie et al, 2015). GO term association and enrichment analysis were performed using the Database for Annotation, Visualization, and Integrated Discovery. The KEGG database (2019-11-19) was used to annotate protein pathways (Kanehisa & Goto, 2000). The KEGG online service tool KAAS was used to annotate the proteins' KEGG database descriptions. The annotation results were mapped on the KEGG pathway database using the KEGG online service tool KEGG Mapper. The domain annotation was performed with InterProScan (version_7.0) on the InterPro domain database via Web-based interfaces and services. WoLF PSORT was used to predict subcellular localization (Horton et al, 2007). The CORUM 3.0 database was used to annotate protein complexes. Motif-X software was used to analyze the models of the sequences with amino acids in specific positions of ubiquityl-21-mers (10 amino acids upstream and downstream of the Kub site) in all of the protein sequences. In addition, the IPI Arabidopsis (*A. thaliana*) proteome was used as the background database, and the other parameters were set to default values. The parameter settings for searching motifs using Motif-X software were occurrences 20 and the Bonferroni-corrected *P* = 0.005. Protein–protein interaction networks were analyzed with the IntAct database (http://www.ebi.ac.uk/intact/). The protein–protein interaction network map was generated with the Cytoscape software (Shannon et al, 2003).

## Data Availability

All datasets have been deposited to the ProteomeXchange Consortium via the PRIDE partner repository with the dataset identifier

PXD027037. The description of dataset was shown in Tables S5, S6, and S7.

# Supplementary Information

# Acknowledgements

We thank the Nottingham Arabidopsis Stock Centre (NASC) for T-DNA insertion lines. Dr. Shi Xiao (SUN YATSEN University) provided the *gin2* and *ppc2* seeds. Dr. Rongcheng Lin (Institute of Botany, Chinese Academy of Sciences) kindly provided the antibody against BRM, and Dr. Keqiang Wu (Taiwan University) provided the *brm* seeds. Dr Wenqiang Tang (Hebei Normal University) provided the *ubp12*, *ubp12ubp13*, and *oeUBP12* seeds. We thank Jingjie PTM Biolabs for providing the methods for partial data analysis. This work was supported by a grant from the National Natural Science Foundation of China (grant number 31770318) and grant of excellent PhD candidate program of Fujian Agriculture and Forestry University (No. 324-1122yb049). We thank Dr. Binghua Wu (Fujian Agriculture and Forestry University) for critically reading the manuscript.

## Author Contributions

W Lan: resources, data curation, software, formal analysis, investigation, and methodology.
W Ma: data curation, formal analysis, investigation, and methodology.
S Zheng: resources, investigation, and methodology.
Y Qiu: resources, investigation, and methodology.
H Zhang: resources, investigation, and methodology.
H Lu: investigation.
Y Zhang: software, formal analysis, validation, and visualization.
Y Miao: conceptualization, data curation, supervision, validation, visualization, project administration, and writing—original draft, review, and editing.

## Conflict of Interest Statement

The authors declare that they have no conflict of interest.

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
