## [Reviewer comments · Life Science Alliance]

Life Science Alliance

Ubiquitome profiling reveals a regulatory pattern of UPL3 with UBP12 on metabolic leaf senescence

Wei Lan, Weibo Ma, Shuai Zheng, Yuhao Qiu, Han Zhang, Haisen Lu, Yu Zhang, and Ying Miao

DOI: <https://doi.org/10.26508/lsa.202201492>

Corresponding author(s): Ying Miao, Fujian Agriculture and Forestry University

Review Timeline:

Submission Date:	2022-04-20
Editorial Decision:	2022-06-02
Revision Received:	2022-06-14
Editorial Decision:	2022-07-15
Revision Received:	2022-07-19
Accepted:	2022-07-19

Scientific Editor: Novella Guidi

Transaction Report:

June 2, 2022

Re: Life Science Alliance manuscript #LSA-2022-01492-T

Prof. Ying Miao
Fujian Agriculture and Forestry University
Cangshanqu, Shangxian Road 15#
Fuzhou, Fujian 350002
China

Dear Dr. Miao,

Thank you for submitting your manuscript entitled "Multi-omics analysis reveals a regulatory pattern of UPL3 on metabolism related leaf senescence" to Life Science Alliance. The manuscript was assessed by expert reviewers, whose comments are appended to this letter. We invite you to submit a revised manuscript addressing the Reviewer comments.

Thank you for this interesting contribution to Life Science Alliance. We are looking forward to receiving your revised manuscript.

Sincerely,

B. MANUSCRIPT ORGANIZATION AND FORMATTING:

Reviewer #1 (Comments to the Authors (Required)):

The manuscript "Multi-omics analysis reveals a regulatory pattern of UPL3 with UBP12 on metabolism related leaf senescence" by Lan et al. is an interesting and generally balanced analysis of the ubiquitin ligase UPL3 and other proteins that most likely interact with UPL3. A particularly interesting aspect is that mutants in ubiquitin protease UBP12 share phenotypes with UPL3 overexpression, and UBP12 overexpression causes phenotypes shared with upl3 mutants. This suggests that the two proteins act antagonistically. Using proteomics and proteomics specifically targeting ubiquitin conjugates, the authors also discover a number of proteins that may be direct targets of UPL3, and many more that are influenced probably indirectly by UPL3. Furthermore, a number of potential interactors of UPL3 are identified.

The manuscript is readable and well prepared. Nonetheless, a few minor changes would increase readability:

The English language is not used without errors. Please double-check.

Line 36, its regulatory mechanism remains limiting: understanding of its regulation remains limited ?

Line 113, less rosette leaf numbers: and lower number of rosette leaves are observed in UPL3 overexpressing plants ?

Line 149, it took me a while to understand what 1/1.5 means; maybe write in parentheses (0.67)

Line 151, Of them means what ?

Line 166 ff and Figure 3A, an increase from 47 to 49% is not large. I would call this "almost no change".

Line 250, algorism: algorithm

Line 331, 333: folds: 1 to 1.5-fold ?

Line 382, the gin2 mutant is not mentioned before. Please state full name of the gene/mutant and a reference.

Line 399 ff, flowering time is usually measured in parallel as number of rosette leaves at the time of bolting, in order to compensate for differences in development of different plant genotypes. Do the authors have these data ? If yes, please include.

Line 466, actin model: action model ?

Line 572, Perhaps it might compete: please correct English

Line 578, 579, showed various of up- or down-regulated: showed various signs of up- ...?

Line 580, it might explain that ...: please correct English

Line 961, complement line: complemented line ?

Line 1029/Fig 5: Panel C please indicate whether the change shown is up or down 1.2 and 1.5-fold, respectively. Panel D, what do the colors mean ? A color ruler is just below, but it probably belongs to panel E. This is confusing.

Fig. 6 panel D lacks a negative control. Please provide (in Supplement ?). Likewise, panel F needs to contain the information how much of the extract was used for the input lane compared to the IP lane, and the total protein control for the input lane.

Fig. 9 is confusing: Red color is used for two protein name abbreviations, color blue only as a background for some boxes, but not for protein names. It would be easier if only blue and red background would be used, or blue and red lettering. Furthermore, what does the brown background color mean ? Just a decorative element ?

Reviewer #2 (Comments to the Authors (Required)):

The manuscript describes the analysis of a UPL ligase protein from HECT E3 protein family using gene mutation, Yeast Two-hybrid screen, label-free proteomics (total and subcellular for ubiquitinated proteins) and qRT-PCR-based gene expression analysis. The authors identified conjugated proteins, potential interacting proteins, reported phenotype alterations in the mutant plants and results of subcellular proteomics, indicating potential role of interacting partner proteins and alterations in the central carbon metabolism and senescence.

The main text needs revision of orthographic errors and misspelling words throughout the whole manuscript (e.g., (conjugates) at legend of Fig 3).

Fig. 1G. Which leaf or leaves was/were selected for the experiments of Fv/Fm measurements? It is important to verify the total content of available chlorophyll and find a proper data normalization, since senescence leaves may not have chlorophyll at all in some areas, therefore reduced Fv/FM values are expected. Maybe normalizing the data by total chlorophyll content may be a way to do that?

The authors mentioned in the abstract that transcriptome analysis was performed but the described method and results regarding gene expression are only related to qRT-PCR analysis. No transcriptome analysis was performed.

There are protein groups reported as valid hits, (ex. F4IGE9) which have no expression value at the proteomics dataset. 1932.

The dataset of proteomics data reports 2887 proteins in total. However, the dataset contains many missing values, and the authors did not discuss this issue and how this affected the statistical analysis.

The reference mentioned as method description for ubiquitinome analysis does not involve a proteome analysis of total proteome, only ubiquitinated proteins. Therefore, the authors should describe in detail how the total proteome data analysis was performed, informing also statistical tests and post hoc tests applied.

The work only presents two replicate datasets from total proteome analysis, which does not allow any statistical conclusion to be addressed from this data. A minimum of three independent experimental replicates of good quality (few non-valid values) must be analyzed to provide strong evidence of their findings. Many statements are done based on this data analysis. Therefore, a more rigorous analysis must be performed since total proteome analysis should not be presented for only two independent experimental replicates. Consequently, the statistic of the results is not strong enough to address too many conclusions.

For ubiquitinated analysis, one of the proteomics samples of mutant strain upl3 (B2) has too many missing values. The authors must explain why this result is considered and why it has so many missing values. Even proteins that have no expression value valid are reported in the spreadsheet. The authors must reanalyze the data considering the need to treat the problem of the number of non-valid values (missing values) and a proper data pre-processing and filtering must be done to provide valid information of proteomics alterations.

The version of the database used for proteome and data annotation must be included in the methods description

The statistical analysis of proteomics data is not completely described. There is no mention about which method was applied for statistic calculation of p-value and if a post-hoc test was performed.

Supp Figure S5. Please include the color scale bar in the image that indicates the significance of the nodes identified.

Supp. Figure S6. Please increase the quality of the image. The words are sometimes not possible to read.

Supp method S2. It is important to add the version of the databases used in the searches and generation of annotation for nodes and pathways.

Supp figure 3C. The results from experimental replicates of samples from the same strain are weakly correlated, achieving the same correlation values shown for non-replicates. The authors must discuss this issue demonstrating why this happens. What were the sources of variation expected? Why so low correlation is observed?

Response to reviewers

Reviewer #1 (Comments to the Authors (Required)):

The manuscript "Multi-omics analysis reveals a regulatory pattern of UPL3 with UBP12 on metabolism related leaf senescence" by Lan et al. is an interesting and generally balanced analysis of the ubiquitin ligase UPL3 and other proteins that most likely interact with UPL3. A particularly interesting aspect is that mutants in ubiquitin protease UBP12 share phenotypes with UPL3 overexpression, and UBP12 overexpression causes phenotypes shared with *upl3* mutants. This suggests that the two proteins act antagonistically. Using proteomics and proteomics specifically targeting ubiquitin conjugates, the authors also discover a number of proteins that may be direct targets of UPL3, and many more that are influenced probably indirectly by UPL3. Furthermore, a number of potential interactors of UPL3 are identified.

The manuscript is readable and well prepared. Nonetheless, a few minor changes would increase readability:

The English language is not used without errors. Please double-check.

Re: Thank you for your efforts for reviewing our manuscript and give positive evaluation. We carefully checked English language through whole manuscript and revised it.

Line 36, its regulatory mechanism remains limiting: understanding of its regulation remains limited ?

Re: corrected

Line 113, less rosette leaf numbers: and lower number of rosette leaves are observed in UPL3 overexpressing plants ?

Re: corrected

Line 149, it took me a while to understand what 1/1.5 means; maybe write in parentheses (0.67)

Re: We revised in the text.

Line 151, Of them means what ?

Re: Of 267 DECs, we corrected in the text.

Line 166 ff and Figure 3A, an increase from 47 to 49% is not large. I would call this "almost no change".

Re: corrected

Line 250, algorism: algorithm

Re: corrected

Line 331, 333: folds: 1 to 1.5-fold ?

Re: corrected

Line 382, the *gin2* mutant is not mentioned before. Please state full name of the gene/mutant and a reference.

Re: we added "a *HXX1* mutant *gin2* (glucose-insensitive2) (Cho et al., 2006)" in the text (line 385).

Line 399 ff, flowering time is usually measured in parallel as number of rosette leaves at the time of bolting, in order to compensate for differences in development of different plant genotypes. Do the authors have these data ? If yes, please include.

Re: yes, the flowering time was shown in the supplementary Fig S1B and Fig S11.

Line 466, actin model: action model ?

Re: corrected

Line 572, Perhaps it might compete: please correct English

Re: corrected

Line 578, 579, showed various of up- or down-regulated: showed various signs of up- ...?

Re: corrected

Line 580, it might explain that ...: please correct English

Re: corrected

Line 961, complement line: complemented line ?

Re: corrected

Line 1029/Fig 5: Panel C please indicate whether the change shown is up or down 1.2 and 1.5-fold, respectively. Panel D, what do the colors mean ? A color ruler is just below, but it probably belongs to panel E. This is confusing.

Re: We added a color ruler in the Fig5D and the description in the legend of Figure 5.

Fig. 6 panel D lacks a negative control. Please provide (in Supplement ?). Likewise, panel F needs to contain the information how much of the extract was used for the input lane compared to the IP lane, and the total protein control for the input lane.

Re: we added the negative controls in Supplementary Fig S8B and CoIP methods was added in the Material and Methods (line 671-675) and the legend of Figure 6F, we add a WT protein control in the new Figure 6F.

Fig. 9 is confusing: Red color is used for two protein name abbreviations, color blue only as a background for some boxes, but not for protein names. It would be easier if only blue and red background would be used, or blue and red lettering. Furthermore, what does the brown background color mean ? Just a decorative element ?

Re: Thanks! We corrected the Figure 9.

Reviewer #2 (Comments to the Authors (Required)):

The manuscript describes the analysis of a UPL ligase protein from HECT E3 protein family using gene mutation, Yeast Two-hybrid screen, label-free proteomics (total and subcellular for ubiquitinated proteins) and qRT-PCR-based gene expression analysis. The authors identified conjugated proteins, potential interacting proteins, reported phenotype alterations in the mutant plants and results of subcellular proteomics, indicating potential role of interacting partner proteins and alterations in the central carbon metabolism and senescence.

The main text needs revision of orthographic errors and misspelling words throughout the whole manuscript (e.g., (conjunates) at legend of Fig 3).

Re: Thank you for your efforts for reviewing our manuscript. We carefully checked English language through whole manuscript and revised it.

The word "conjunates" is corrected.

Fig. 1G. Which leaf or leaves was/were selected for the experiments of Fv/Fm measurements? It is important to verify the total content of available chlorophyll and find a proper data normalization, since senescence leaves may not have chlorophyll at all in some areas, therefore reduced Fv/FM values are expected. Maybe normalizing the data by total chlorophyll content may be a way to do that?

Re: The 5th of rosette leaf of 6-week-old plants was used for Fv/Fm measurement. At this stage, total chlorophyll content and Fv/Fm value has no significantly different. It is a good question. What is an early senescence marker? Actually, senescence is initiated before chlorophyll degradation. We added the data of transcript level of early senescence marker gene *WRKY53* (Miao et al., 2004) and floral transition gene *FT* in the Supplementary Fig S2D and the description in the text (line 121-124).

The authors mentioned in the abstract that transcriptome analysis was performed but the described method and results regarding gene expression are only related to qRT-PCR analysis. No transcriptome analysis was performed.

Re: We revised the description in the text.

There are protein groups reported as valid hits, (ex. F4IGE9) which have no expression value at the proteomics dataset. 1932.

Re: Yes, due to the method limitation, the variant of three replicates proteomics dataset is quite high. We collected the protein values at least appeared in two replicates to analyze. We did not show more detail results of proteome dataset. Proteome data analysis is not main point for this manuscript, because overlapping analysis of DUCs of ubiquitome dataset and DEPs of proteome dataset showed a few proteins were overlapped and changed between two datasets, the interested proteins had been confirmed by western blotting (Figure 7), confirming that UPL3 function are not directly act in UPS mediating protein degradation. The altered proteins of proteome dataset are mainly involved in UPL3-regulated downstream function. We uploaded DEPs in the Supplementary dataset S4. We changed the title to "Ubiquitome profiling reveals a regulatory pattern of UPL3 with UBP12 on metabolism related leaf senescence ". We added the discussion in the text (476-491).

The dataset of proteomics data reports 2887 proteins in total. However, the dataset contains many missing values, and the authors did not discuss this issue and how this affected the statistical analysis.

Re: We did three replicates, However, the variant of three replicate datasets are quite high. We collected the site (protein) values at least appeared in two replicates to analyze. We added the discussion in the text (476-491). The most of sites (proteins) missed values were a few sites of protein, perhaps it has special meaning under some condition. We kept the original dataset in the Supplementary dataset S1.

The reference mentioned as method description for ubiquitinome analysis does not involve a proteome analysis of total proteome, only ubiquitinated proteins. Therefore, the authors should describe in detail how the total proteome data analysis was performed, informing also statistical teste and post hoc tests applied.

Re: we added the description of proteome analysis in the Supplementary Method S2. We collected the protein values at least appeared in two replicates to analyze. Proteome data analysis is not main point for this manuscript, we need to add more replicate datasets and describe in detail how the total proteome data analysis was performed in next manuscript.

The work only presents two replicate datasets from total proteome analysis, which does not allow any statistical conclusion to be addressed from this data. A minimum of three independent experimental replicates of good quality (few non-valid values) must be analyzed to provide strong evidence of their findings. Many statements are done based on this data analysis. Therefore, a more rigorous analysis must be performed since total proteome analysis should not be presented for only two independent experimental replicates. Consequently, the statistic of the results is not strong enough to address too many conclusions.

Re: You are right. We did three replicates, However, the variant of three replicate datasets are quite high. We collected the protein values at least appeared in two replicates to analyze. We have done masses of works for protein-protein interaction screening and identifying the candidates. Proteome data analysis is not main point for this manuscript. It is used to confirm few of candidates for ubiquitome datasets. The interested proteins had been confirmed by western blotting (Figure 7) and phenotyping, confirming that UPL3 function are not directly act in UPS mediating protein degradation. We changed the title of manuscript "Ubiquitome profiling reveals a regulatory pattern of UPL3 with UBP12 on metabolism related leaf senescence ". In my opinion, our evidence can support our conclusion. We discussed it in the discussion part.

For ubiquitinated analysis, one of the proteomics samples of mutant strain upl3 (B2) has too many missing values. The authors must explain why this result is considered and why it has so many missing values. Even proteins that have no expression value valid are reported in the spreadsheet. The authors must reanalyze the data considering the need to treat the problem of the number of non-valid values (missing values) and a proper data pre-processing and filtering must be done to provide valid information of proteomics alterations.

The version of the database used for proteome and data annotation must be included in the methods description

Re: We did three replicates, However, the variant of three replicate datasets are quite high. due to the method limitation. We collected the protein values at least appeared in two replicates to analyze, the rest dataset contains many missing values. Afterward, we did masses of works to screen and identify the candidates, the interested proteins had been confirmed by western blotting and phenotyping. We have added the DUCs of ubiquitome dataset and DEPs of proteome dataset in the Supplementary datasets.

The statistical analysis of proteomics data is not completely described. There is no mention about which method was applied for statistic calculation of p-value and if a post-hoc test was performed.

Re: We added the method was applied for statistic calculation of p-value in the Supplementary Method 2. Proteome data analysis is not main point for this manuscript, we need to add more replicate datasets and describe in detail how the total proteome data

analysis was performed in next manuscript.

Supp Figure S5. Please include the color scale bar in the image that indicates the significance of the nodes identified.

Re: corrected. We added the indications in the legend of Fig S5

Supp. Figure S6. Please increase the quality of the image. The words are sometimes not possible to read.

Re: corrected

Supp method S2. It is important to add the version of the databases used in the searches and generation of annotation for nodes and pathways.

Re: done

Supp figure 3C. The results from experimental replicates of samples from the same strain are weakly correlated, achieving the same correlation values shown for non-replicates. The authors must discuss this issue demonstrating why this happens. What were the sources of variation expected? Why so low correlation is observed?

Re: We did three replicates, however, the variant of three replicate datasets are quite high, due to the method limitation. Afterward, we have done masses of works to screen and identify the candidates. we discussed it in the discussion section (line 474-488).

July 15, 2022

RE: Life Science Alliance Manuscript #LSA-2022-01492-TR

Prof. Ying Miao
Fujian Agriculture and Forestry University
Cangshanqu, Shangxian Road 15#
Fuzhou, Fujian 350002
China

Dear Dr. Miao,

Thank you for submitting your revised manuscript entitled "Ubiquitome profiling reveals a regulatory pattern of UPL3 on metabolism related leaf senescence". We would be happy to publish your paper in Life Science Alliance pending final revisions necessary to meet our formatting guidelines.

- please consult our manuscript preparation guidelines <https://www.life-science-alliance.org/manuscript-prep> and make sure your manuscript sections are in the correct order
- please upload supplementary figures as single files
- please upload your table files as editable excel or doc files
- please add a section to your manuscript for the main and supplementary figures and the tables
- please add the Twitter handle of your host institute/organization as well as your own or/and one of the authors in our system
- please use the [10 author names, et al.] format in your references (i.e. limit the author names to the first 10)
- please add a callout for Figure S2A,B; Figure S12A; you have a callout for Figure 4E in the manuscript, but this doesn't exist in the figure or the legend
- please integrate your supplementary methods in the main Materials and Methods section

Figure Check:

- please add a panel C to your Figure S1-it is in the legend for Figure S1 but not in the figure itself
- please add a panel C to your figure S8 legend
- Figure 1B needs scale bars
- Figure 2C bottom row looks like a splice after the 2nd blot. Please provide source data.
- Figure 3F needs scale bars
- Figure 5A right section: looks like a splice before the last column of blots. Please provide source data.
- Fig. 6B,C: scale bars needed; Fig. 6D,E: scale bars needed
- Fig. 6F top row: duplicates of 1st and 3rd sections; splice between blots in bottom row, 2nd part; please provide source data.
- Figure 7D: splices throughout the figure, please provide source data.
- Figure 7E, top right and 2nd row: looks like blots have been copied in; middle left: splice after 2nd blot; 3rd, 4th row left: middle looks like a copied in blank lane; right part of 3rd and 4th row also look like there's a copied in lane and clear splice after 3rd blot of bottom right row. Please provide source data.
- Figure 8B: scale bars needed
- Figure S2B: scale bars needed
- Figure S8A,B: scale bars needed

A. FINAL FILES:

B. MANUSCRIPT ORGANIZATION AND FORMATTING:

Sincerely,

Reviewer #1 (Comments to the Authors (Required)):

The authors have sufficiently taken reviewer suggestions into account.

July 19, 2022

RE: Life Science Alliance Manuscript #LSA-2022-01492-TRR

Prof. Ying Miao
Fujian Agriculture and Forestry University
Cangshanqu, Shangxian Road 15#
Fuzhou, Fujian 350002
China

Dear Dr. Miao,

Thank you for submitting your Research Article entitled "Ubiquitome profiling reveals a regulatory pattern of UPL3 with UBP12 on metabolic leaf senescence". It is a pleasure to let you know that your manuscript is now accepted for publication in Life Science Alliance. Congratulations on this interesting work.

DISTRIBUTION OF MATERIALS:

Again, congratulations on a very nice paper. I hope you found the review process to be constructive and are pleased with how the manuscript was handled editorially. We look forward to future exciting submissions from your lab.

Sincerely,
